# SPDEBench: An Extensive Benchmark for Learning Stochastic PDEs

## Abstract

Stochastic Partial Differential Equations (SPDEs) driven by random noise play a central role in modeling physical processes with rough spatio-temporal dynamics, such as turbulence flows, superconductors and quantum dynamics. To efficiently model these processes and make predictions, machine learning (ML)-based surrogate models are proposed, with spatio-temporal roughness incorporated in the design of their network architectures. However, an extensive and unified dataset for SPDE learning is still lacking; in particular, existing datasets do not account for the computational error introduced by noise sampling and the necessary renormalization required for handling singular SPDEs. We thus introduce SPDEBench, aiming to solve typical SPDEs of physical significance on 1D or 2D tori driven by white noise via ML methods. New datasets for singular SPDEs based on the renormalization process, as well as novel ML models achieving the best results to date, have been proposed. Moreover, we evaluate the sensitivity of ML models to the SPDE data generation setting and the hyperparameters, and investigate the scaling law of ML models with respect to sample and network sizes. Results are benchmarked with ML-based models, including FNO, NSPDE and DLR-Net, etc. By evaluating performance from multiple perspectives, we achieve a comprehensive assessment of the relative strengths and weaknesses of different ML models. Our SPDEBench ensures full reproducibility of benchmarking across a variety of SPDE datasets while offering the flexibility of incorporating new datasets and machine learning baselines, thus making it a valuable resource for the community.[1]

## 1 Introduction

Stochastic Partial Differential Equations (SPDEs) driven by random noise are important mathematical models for random processes; *e.g.*, turbulence flows, superconductors, quantum dynamics, filtering in finance, and real-time control Hairer (2013; 2014); Klauder (1983); Peng (1992). SPDEs are both of physical significance and have triggered notable advances in theoretical mathematics. For example, one major breakthrough in the analysis of SPDEs in this decade is Fields Medalist M. Hairer's theory of regularity structures Hairer (2014). The SPDE learning, *i.e.*, using machine learning (ML) models to approximate SPDE dynamics, also attracts a lot of attention Chevyrev et al. (2024); Salvi et al. (2022); Gong et al. (2023); Neufeld & Schmocker (2024). One promising framework involves modeling the temporal evolution of parametric SPDEs via deep neural networks: the neural network learns a mapping from input functions (*e.g.*, initial conditions and noise paths) to the corresponding solutions Gong et al. (2023); Salvi et al. (2022); Li et al. (2020). ML-based models such as Neural SPDE Salvi et al. (2022) and DLR-Net Gong et al. (2023) not only accelerate future state prediction of certain SPDEs, but also lead to novel network architectures — which incorporate physical prior in better ways — for spatio-temporal dynamics modeling.

Despite the importance of SPDEs, the field of machine learning SPDEs is still far from being fully explored. First, existing datasets for learning SPDEs are limited, numerical simulation for SPDEs is highly specialized, and off-the-shelf software for such simulations is lacking. For example, in the theory of regularity structures Hairer (2013), solutions for singular SPDEs are obtained via a delicate renormalization approach (i.e., first considering a sequence of regularized problems driven by mollifications (*i.e.*, the natural smoothing) of the singular noise $\xi$, and then subtracting a large

---

[1]The code will be made publicly available upon the paper's publication.

constants ("infinities") from the regularized solutions), but datasets for learning singular SPDEs with renormalization are completely missing. Second, many studies of ML-based models for learning SPDEs overlook important evaluation metrics and/or parameters that are important for the numerical SPDE theory (*e.g.*, the approximation error caused by renormalization and noise truncation Lord et al. (2014)), hence leading to biased and incomplete evaluations of the ML-models.

To address these challenges, we propose a unified benchmark for studying ML-based models for SPDEs, aiming to facilitate and accelerate the advancement in designing effective and robust ML-based models for approximating dynamics driven by SPDEs (*e.g.*, the $\Phi_d^4$, wave, incompressible Navier–Stokes, KPZ and KdV equations). These SPDEs are mathematically well studied, hold fundamental significance in statistical physics and fluid mechanics and encapsulate core challenges like non-convex potentials (e.g., $\Phi_d^4$, KPZ, incompressible Navier–Stokes equations) and soliton dynamics (e.g., KdV and stochastic wave). We introduce **SPDEBench**, the first benchmark for learning SPDEs which can be used to numerically generate solutions of various SPDEs, pre-generated ready-to-use datasets which cover well-known singular and nonsingular SPDEs, multiple evaluation metrics for training and comparing the ML models, as well as evaluation results of ML models like FNO Li et al. (2020), NSPDE and DLR-Net for learning SPDEs. SPDEBench also serves as a controlled testbed for evaluating ML algorithms on empirical spatio-temporal datasets whose evolution is traditionally modeled by SPDEs.

SPDEBench has the following distinct features: (a) **Data generation:** it takes into consideration the renormalization procedure and the approximation error in the numerical computation of random noise, thus enabling data generation under various settings of noise truncation degree or renormalization parameters in a user-friendly way; (b) **Evaluation:** it provides testing metrics of the machine learning model beyond RMSE, e.g., $W^{1,2}$ sobolev norm, correlation score, scaling law, etc; (c) **Model comparasion findings:** incorporating renormalization constant or the regularity feature inspired by the regularity structure theory into the design of the neural networks can enhance the model's accuracy and robustness to the noise truncation degree and evaluation metrics.

## 2 PRELIMINARIES

In this section, we introduce the fundamental concepts of SPDEs and outline the general formulation for SPDE learning tasks.

### 2.1 MILD SOLUTION OF SPDE

We consider SPDEs of the form:

$$\begin{aligned}
\partial_t u - \mathcal{L}u &= \mu(u, \partial_1 u, \cdots, \partial_d u) + \sigma(u, \partial_1 u, \cdots, \partial_d u)\xi \qquad \text{in } [0, T] \times D, \\
u(0, x) &= u_0(x),
\end{aligned} \tag{1}$$

where $x \in D \subset \mathbb{R}^d$, $t \in [0, T]$, $\partial_i := \partial/\partial x_i$, $i \in \{1, \ldots, d\}$, $\mathcal{L}$ is a linear differential operator, $\xi : [0, T] \times D \to \mathbb{R}$ is the stochastic forcing, $u_0 : D \to \mathbb{R}$ is the initial datum, and $\mu, \sigma : \mathbb{R} \times \mathbb{R}^d \to \mathbb{R}$ are given functions. Under suitable conditions on $\mu, \sigma$ (*e.g.*, locally Lipschitz), this SPDE has a unique *mild solution* representable via the operator semigroup $\{e^{t\mathcal{L}}\}$ Hairer (2013); Salvi et al. (2022):

$$u_t = e^{t\mathcal{L}}u_0 + \int_0^t e^{(t-s)\mathcal{L}}\mu(u_s, \partial_1 u_s, \cdots, \partial_d u_s)\,\mathrm{d}s + \int_0^t e^{(t-s)\mathcal{L}}\sigma(u_s, \partial_1 u_s, \cdots, \partial_d u_s)\xi(\mathrm{d}s),$$

where $u_t(\cdot) := u(t, \cdot); t \in [0, T]$. In general, if $\xi = \xi(t, x)$ is space-time white noise, SPDEs have singularity in space and time. It is highly irregular: for parabolic equations it lies in the negative Hölder space $\mathcal{C}^{-\frac{d+2}{2}-}$ in spatial dimension $d = 1, 2, 3, \ldots$ Gubinelli et al. (2015); Hairer (2013). Formal definition of space-time white noise can be found in Appendix C.1.

### 2.2 CYLINDRICAL WIENER PROCESS

Most of the interesting SPDEs do not admit explicit analytical solutions, so the study of computational SPDEs is of great practical and theoretical importance Lord et al. (2014). The deterministic part of SPDEs can be discretized or approximated through traditional numerical methods; *e.g.*, finite difference/finite element methods. Thus, in the sequel we focus on the computation of the stochastic

forcing $\xi$. In all SPDEs in this benchmark except Navier-Stokes, $\xi$ is the space-time white noise, which is equivalent to the time derivative of the cylindrical Wiener process.

**Definition 2.1.** Da Prato & Zabczyk (2014)[Cylindrical Wiener process] Let $H$ be a separable Hilbert space. A cylindrical Wiener process is the following $H$-valued stochastic process: $W(t) = \sum_{j=1}^{\infty} \phi_j \beta_j(t), t \geq 0$, where $\{\phi_j\}$ is any orthonormal basis (ONB) for $H$ and $\beta_j(t)$ are i.i.d. Brownian motions.

Cylindrical Wiener process can be regarded as a special case of Q-Wiener process (see details in Appendix C.2). In this benchmark, we provide different realizations of $\xi$ in terms of the noise type (i.e., cylindrical Winner process or Q-Wiener process), two basis choices for the Hilbert space (i.e., Fourier basis or Haar wavelet basis), and different truncation degrees of the summation in the representation of $W(t)$. For example, in 1D case, we set the spatial domain $D = (0, L)$ and consider an $L^2(D)$-valued cylindrical Wiener process $W(t)$, which is represented by the Fourier basis or Haar wavelet basis. We take the ONB $\phi_k(x) = \sqrt{2/L} \sin\left(k\pi x/L\right)$ and sample from the truncated expansion with degree $J$, i.e., $W^J(t) = \sum_{j=1}^{J} \phi_j \beta_j(t)$ at the points $x_m = {}^{mL}/_N$ for $m = 1, \ldots, N$. In 2D case, we set $D = (0, L_x) \times (0, L_y)$, take the $L^2(D)$-ONB $\phi_{j,k}(x, y) = \sqrt{4/L_x L_y} \sin\left(j\pi x/L_x\right) \sin\left(k\pi y/L_y\right)$, and sample from the truncations $W^{J_x, J_y}(t) = \sum_{j=1}^{J_x} \sum_{k=1}^{J_y} \phi_{j,k} \beta_{j,k}(t)$ at $x_m = {}^{mL_x}/_{N_x}, y_n = {}^{nL_y}/_{N_y}$ for $m = 1, \ldots, N_x; n = 1, \ldots, N_y$.

## 2.3 FORMULATION OF THE LEARNING PROBLEM

The objective of machine learning (ML) tasks for SPDE is to find some ML-based surrogate to approximate the mild solution $u_t(x)$ of the SPDEs. The solution depends not only on the previous time steps of the solution, but also on the random forcing. As this work mainly focused on how the singularity with respect to the random force influences the learning, the learning task is to approximate the following operators:

- $\mathcal{G}_1 : \xi_t(x) \to u_t(x), t \in (0, T]$ under fixed initial datum $u_0(x)$;
- $\mathcal{G}_2 : (u_0(x), \xi_t(x)) \to u_t(x), t \in (0, T]$ where $u_0(x)$ follows a given distribution.

In practice, the temporal and spatial domain gets discretized into meshes and $u_t(x), x, t \in mesh$ is approximated by numerical methods. The target now becomes approximating the following operators using ML-based surrogates:

- $\hat{\mathcal{G}}_1 : \{\hat{\xi}_t(x); x, t \in mesh\} \to \{\hat{u}_t(x); x, t \in mesh\}$ under fixed intial condition $u_0(x)$;
- $\hat{\mathcal{G}}_2 : \{u_0(x), \hat{\xi}_t(x); x, t \in mesh\} \to \{\hat{u}_t(x); x, t \in mesh\}$ where $u_0(x)$ follows a given distribution.

Here, $\hat{\xi}_t(x)$ is sampled by the cylindrical Wiener process (or Q-Wiener process) with truncation degree $J$, and $\hat{u}_t(x)$ is generated by numerical solvers. Given a dataset that consists $n$ sample trajectories $\mathcal{Z} = \left\{ \left( \hat{\xi}^{(i)}; \hat{u}^{(i)} \right) \middle| i = 1, \cdots, n; u_0; J \right\}$ or $\mathcal{Z} = \left\{ \left( u_0^{(i)}, \hat{\xi}^{(i)}; \hat{u}^{(i)} \right) \middle| i = 1, \cdots, n; J \right\}$, the ML-based models, e.g., the neural network $f_\theta$, aim to achieve a sample-wise path-to-path fit between the true and estimated solutions of the SPDE. This is typically achieved by solving the following optimization problem that minimizes the discrepancy between them.

$$f^* = \arg\min_\theta \frac{1}{n} \sum_{i=1}^{n} L\left( f_\theta\left( \hat{\xi}^{(i)}, u_0^{(i)} \right), \hat{u}^{(i)} \right), \tag{2}$$

where $L(\cdot, \cdot)$ denotes the loss function. A commonly used loss function is the relative $L^2$-loss Li et al. (2020); Salvi et al. (2022).

## 3 SPDEBENCH: A BENCHMARK FOR LEARNING SPDEs

In the following, we describe the details of SPDEBench, including the overview of datasets, existing implemented ML models developed using PyTorch, and implementation guidance for users.

## 3.1 OVERVIEW OF DATASETS

We are primarily concerned with the SPDEs, which are broadly representative and fundamental to statistical physics and fluid mechanics. In the SPDEBench, we generate two classes of the datasets: the nonsingular SPDEs and the singular SPDEs datasets. The key information of the SPDEs in SPDEBench is summarized in Table 1 and Table 2. We generate datasets under various configurations of numerical solver, including the noise type of $W(t)$, the choice of basis functions $\phi_j$, the noise truncation degree $J$ and whether renormalization is applied during data generation.

Table 1: SPDEs with spatial domain $D$, time interval $[0, T]$, spatial resolution $N_D$, temporal resolution $N_t$, total number of generated samples $N_{\mathcal{Z}}$ (i.e., $c * z$ means we tried $c$ distinct configurations in the numerical solver, with $z$ samples generated for each). The rightarrow $n \to m$ means the resolution is downsampled from $n$ to $m$. For the dynamic $\Phi_2^4$ model, the notation "–S" (or "–L") denotes datasets containing 1200 (or 10000) number of samples, respectively.

| SPDE | Space | Time | $N_D$ | $N_t$ | $N_{\mathcal{Z}}$ | Singular |
|---|---|---|---|---|---|---|
| Dynamic $\Phi_1^4$ | $[0,1]$ | $[0,0.05]$ | 128 | 50 | 8*10000 | No |
| Korteweg–De Vries | $[0,1]$ | $[0,0.5]$ | 128 | 50 | 4*10000 | No |
| Wave | $[0,1]$ | $[0,0.5]$ | 128 | $500 \to 100$ | 4*10000 | No |
| Incompressible NSE | $[0,1]^2$ | $[0,1]$ | $64^2 \to 16^2$ | $1000 \to 100$ | 5*10000 | No |
| Dynamic $\Phi_2^4$-S | $[0,1]^2$ | $[0,0.025]$ | $32 \times 32$ | 250 | 8*1200 | Yes |
| Dynamic $\Phi_2^4$-L | $[0,1]^2$ | $[0,0.025]$ | $32 \times 32$ | 250 | 2*10000 | Yes |
| Kardar–Parisi–Zhang | $[0,1]$ | $[0,0.05]$ | 128 | 50 | 16*10000 | Yes |

Table 2: Configurations of numerical solver used in datasets generation.

| SPDE | Noise type | Basis function | $J$ | Renorm |
|---|---|---|---|---|
| Dynamic $\Phi_1^4$–F | cylindrical Wiener | Fourier | $\{32, 64, 128, 256\}$ | No |
| Dynamic $\Phi_1^4$–H | cylindrical Wiener | Haar wavelet | $\{32, 64, 128, 256\}$ | No |
| Korteweg–De Vries–$cyl$ | cylindrical Wiener | Fourier | $\{32, 64, 128, 256\}$ | No |
| Korteweg–De Vries–Q | Q-Wiener | Fourier | $\{32, 64, 128, 256\}$ | No |
| Wave | cylindrical Wiener | Fourier | $\{32, 64, 128, 256\}$ | No |
| Incompressible NSE | Q-Wiener | Fourier | $\{32, 64, 128, 256\}$ | No |
| Dynamic $\Phi_2^4$–S | cylindrical Wiener | Fourier | $\{2, 8, 32, 64, 128\}$ | $\{Yes,No\}$ |
| Dynamic $\Phi_2^4$–L | cylindrical Wiener | Fourier | $\{128\}$ | $\{Yes,No\}$ |
| Kardar–Parisi–Zhang | cylindrical Wiener | Fourier | $\{32, 64, 128, 256\}$ | Yes |

### 3.1.1 NONSINGULAR SPDE DATASETS

The nonsingular SPDE datasets include 1D Dynamic $\Phi_1^4$ Model, 1D Korteweg–De Vries (KDV) equation, 1D wave equation and 2D incompressible Navier–Stokes equation (NSE, in the form of vorticity equation). For each equation, we follow the data simulation approaches of Salvi et al. (2022); Chevyrev et al. (2024) and generate solution paths by employing four or five settings on the noise truncation degree $J$ (range from 32 to 256). In Table 1 and Table 2, we list the main hyperparameters and configurations in the numerical solvers to generate these datasets. Due to the overlapping content with the references, full details are provided in the Appendix C.4.

### 3.1.2 SINGULAR SPDE DATASETS

In this section, we introduce two datasets for singular SPDEs: the dynamic $\Phi_2^4$ model and the Kardar–Parisi–Zhang (KPZ) equation, whose data generation requires the renormalization procedure. The domains of these equations are the torus $\mathbf{T}^1 = \mathbb{R}/\mathbb{Z}$ obtained by identifying the endpoints of the interval $[0, 1]$ for 1D, and $\mathbf{T}^2$ obtained by identifying the opposite sides of the square $[0, 1]^2$ for 2D.

**The dynamic $\Phi_2^4$ model.** Consider the dynamic $\Phi_2^4$ model:

$$\begin{cases} \partial_t u = \Delta u - u^3 + \sigma\xi & \text{in } [0,T] \times \mathbf{T}^2, \\ u\big|_{t=0} = u_0 & \text{at } \{0\} \times \mathbf{T}^2. \end{cases} \tag{3}$$

We set the initial condition as $u_0(x,y) = \sin(2\pi(x+y)) + \cos(2\pi(x+y)) + \kappa\eta(x,y)$ with $\eta(x,y) = a_0 + \sum_{j=-10}^{j=10}\sum_{k=-10}^{k=10} \frac{a_{j,k}}{|j|^2+|k|^2+1}\sin((j\pi x - k\pi y)/2)$, where $a_0, a_{j,k} \overset{\text{i.i.d.}}{\sim} \mathcal{N}(0,1)$. The numerical solution is calculated with the temporally 1st order and spatially 2nd order difference scheme. The space-time white noise is sampled from the truncated cylindrical Wiener process and scaled by $\sigma = 0.1$. In this case, the nonlinear term $u^3$ is undefined in classical sense (see details in Appendix C.3). Its rigorous definition requires the renormalization procedure, which shall be implemented in this work via the following steps.

**Wick power of stochastic convolution.** Firstly, consider the stochastic convolution $X$: $\partial_t X - \Delta X = \xi$, $X(0) = u_0$, and replace $\xi$ with a suitable mollification $\xi_\varepsilon = \xi * \eta_\varepsilon$ (where $\eta_\varepsilon$ stands for a smoothing kernel). For notational convenience, we rewrite the stochastic convolution with truncated Galerkin projection and solve for $X_\epsilon := X_{J^{-1}}$ from $\mathrm{d}X_{J^{-1}} - \Delta X_{J^{-1}}\,\mathrm{d}t = \mathrm{d}W^J(t)$. The *Wick power* of $X$, denoted as $X^{\diamond n}$ ($n = 2, 3$), is defined as

$$X^{\diamond 2} := \lim_{J \to \infty}\left(X_{J^{-1}}^2 - a_{J^{-1}}\right), \qquad X^{\diamond 3} := \lim_{J \to \infty}\left(X_{J^{-1}}^3 - 3a_{J^{-1}}X_{J^{-1}}\right),$$

with $a_\epsilon := a_{J^{-1}} = \mathbb{E}\left[X_{J^{-1}}^2\right]$ Gubinelli & Hofmanová (2019). In the numerical computation, we use the full training samples to approximately calculate $a_\epsilon$. Now we substitute the ansatz $u(t) = X(t) + v(t)$ into the dynamic $\Phi_2^4$ model, with $X$ is the stochastic convolution above. Then $v$ satisfies the shift equation: $\mathrm{d}v = \Delta v\,\mathrm{d}t - u^3\,\mathrm{d}t, v(0) = 0$, where $u^3$ should be interpreted as the renormalization term $u^{\diamond 3}$ given by the Wick powers of $X$: $u^{\diamond 3} \equiv v^3 + 3v^2 X + 3v X^{\diamond 2} + X^{\diamond 3}$. The remornalized equation is now well-defined:

$$\partial_t u - \Delta u + u^{\diamond 3} = \xi, \quad u(0) = u_0. \tag{4}$$

Then, we implement 2nd order finite diference method (central difference scheme) to numerically calculate Equation (4), and the response paths are generated using $32 \times 32$ spatial resolution and 250 evenly distanced temporal grid point. Figure C1 in Appendix shows the comparison of two data generation methods of $\Phi_2^4$ when truncation degree $J = 128$. It shows that the averaged solution generated by numerically solve the renormalized equation is less noisy and has clearer pattern.

**Kardar–Parisi–Zhang (KPZ) equation.** Consider the KPZ equation:

$$\begin{cases} \partial_t h = \partial_x^2 h + \lambda(\partial_x h)^2 + \xi & \text{in } [0,T] \times \mathbf{T}, \\ h\big|_{t=0} = h_0 & \text{at } \{0\} \times \mathbf{T}. \end{cases} \tag{5}$$

Here, $h(x,t)$ is a continuous stochastic process with $x \in \mathbf{T}$, $\lambda > 0$ is a "coupling strength" and $\xi$ denotes space-time white noise. We set the initial condition as $h_0(x) = \sin(2\pi x) + \cos(2\pi x) + \kappa\eta(x)$ with $\eta(x) = \sum_{k=-10}^{k=10} \frac{a_k}{(|k|+1)^2}\sin(2k\pi x)$, where $a_k \overset{\text{i.i.d.}}{\sim} \mathcal{N}(0,1)$. The nonlinear term $(\partial_x h)^2$ remains undefined in calssical sense, an "infinite constant" is required to renormalize the divergence. According to section 2.1 from Hairer (2013), we consider the following equation

$$dh = \partial_x^2 h dt + \lambda(\partial_x h)^2 dt - \lambda C_\varepsilon dt + dW^\varepsilon(t), \tag{6}$$

where $W^\varepsilon(t) = \sum_{j=1}^\infty \varphi(j\varepsilon)\phi_j\beta_j(t)$, $C_\varepsilon = \frac{1}{\varepsilon}\int_{\mathbf{R}}\varphi^2(x)dx$ with the constant $R$ selected so that $\int_{\mathbf{R}}\varphi^2(x)dx = 1$, $\varphi(x)$ is defined as $\varphi(x) = \exp(1 - \frac{1}{1-(\frac{x}{R})^2})$ when $|x| < R$, otherwise $\varphi(x) = 0$. Then, we set $J = \frac{1}{\varepsilon}$ and choose $\varphi$ such that $C_\varepsilon = J$. We use $W^J$ as an approximation for $W^\varepsilon$. This is the usual way to renormalize the KPZ equation induced by the Cole–Hopf transform. We implement 2nd order finite difference method (central difference scheme) to numerically calculate Equation (6), and the response paths are generated using 128 spatial resolution and 50 evenly distanced temporal grid point. Figure C2 in Appendix shows that the solution generated without implementing the renormalization grows significantly faster over time, displaying a tendency to blow up.

## 3.2 BASELINE ML SURROGATE MODELS

SPDEBench implements several network architectures as surrogate models for practical use and comparative analysis. SPDEBench includes the following existing baseline models and also a modified model for learning singular SPDE.

**NCDE, NRDE and NCDE-FNO** These three rough path theory-inspired ML models Kidger et al. (2020); Morrill et al. (2021); Salvi et al. (2022) incorporate the signatures of controlled differential equations into the network architecture design. They are particularly suitable for irregularly sampled time-series data, allowing for adaptive computation and memory-efficient training.

**FNO, Wavelet Operator and DeepONet** Fourier neural operator Li et al. (2020), Wavelet neural operator Tripura & Chakraborty (2023) and DeepONet Lu et al. (2021) are three representative Neural Operators designed for approximating solutions of parametric PDEs. They are capable of modelling maps between function spaces, which serve as generalizations for classical neural networks.

**NSPDEs and NSPDE-S** Taking spatial-temporal randomness into account, Salvi *et al*Salvi et al. (2022) introduced neural SPDE (NSPDE), a neural operator architecture for modeling operators in SPDEs that take both the initial data and stochastic forcing as inputs. For the NSPDE-S, we incorporate the normalized renormalization constant $a_\epsilon$ computed from the input noise trajectory as a scaling factor by multiplying it with the latent space variables in the NSPDE model.

**DLR-Net** Deep Latent Regularity Network (DLR-Net) Gong et al. (2023) is a neural architecture to combine the regularity feature in the theory of regularity structures with deep neural network, making it particularly robust for SPDE learning whose solution may be rough both temporally and spatially.

## 3.3 EVALUATION METRICS

The standard metric for evaluating ML surrogate models is the relative $L^2$-error (RMSE) on test data. However, it fails to capture small spatial-scale changes that are critical in physical regimes. We also include the following metrics: (1) nRMSE (normalized RMSE) and maximum error over time steps Takamoto et al. (2022); (2) $W^{p,q}$-Sobolev norm (i.e., $\|u\|_{W^{p,q}(\Omega)} = \left(\sum_{|\alpha|\leq p} \|D^\alpha u\|_{L^q(\Omega)}^q\right)^{1/q}$) of the sample-wised difference between the true and the estimated SPDE solutions; (3) correlation score; a statistics-based metric to assess the spatial fitting that measure the $L^2$-difference of the correlation of $(X_t(x), X_t(y))$ between the true and estimated SPDE solutions; (4) autocorrelation score, a statistics-based metric to assess the temporal fitting that measures the $L^2$-difference in the autocorrelation between the true and estimated SPDE solutions.

## 3.4 DATA FORMAT

The benchmark comprises multiple data files in Parquet format, each corresponding to a specific combination of equation, initial condition type, noise type, truncation degree of the driving noise, and data generation method. File naming format is {SPDE name}-{Tasks}-{Truncation degree}-{Sample size}.parquet. For the $\Phi_1^4$ equation, different value of the parameter $\sigma$ are denoted by *01* and *1*. For the $\Phi_2^4$ equation, data generated with renormalization and without renormalization are denoted by *reno* and *expl*, respectively. Each file contains multiple one-dimensional arrays of length $N \times T \times X \times Y$, generated by flattening arrays with dimensions $N, T, X, Y$ with $N$ the number of samples, $T$ the number of time steps, and $X, Y$ the spatial dimensions. While our implementation of SPDEBench for the non-singular SPDEs builds upon the GitHub repositories of Neural SPDE Salvi et al. (2022) and DLR-Net Gong et al. (2023),it significantly extends them by providing a unified and modular benchmarking framework that supports a broader range of ML-based surrogate SPDE models for comprehensive evaluation. It should be noted that, to construct the Parquet-formatted data, all data have been flattened into 1D arrays. When accessing the data, users will need to reshape these arrays back to their original dimensions. We provide detailed code for this process in Appendix D.

## 4 EXPERIMENTS

In this section, we explore two learning tasks introduced in Section 2.3 and denote the two tasks as $\xi \to u$ and $(u_0, \xi) \to u$, respectively. In all the experiments in this section, the training function is selected as the relative $L^2$-error. We present an extensive set of experiments on the $\Phi^4$ datasets and put additional experiments on other datasets, details on hyper-parameters, evaluation with statistical significance of ML surrogate models can be found in Appendix F.

Table 3: Relative $L^2$-error on the test set of the $\Phi_1^4$. Data is generated with different truncation degree ($J = 32|64|128|256$) and $\sigma = 0.1$.

| Model | #Para | Inference time (ms) | $\xi \mapsto u$ $J = 32|64|128|256$ | $(u_0, \xi) \mapsto u$ $J = 32|64|128|256$ |
|---|---|---|---|---|
| Solver | x | 2.438 | x | x |
| NCDE | 545088 | 0.197 | 0.063|0.064|0.067|0.099 | 0.103|0.106|0.109|0.147 |
| NRDE | 8656656 | 0.201 | 0.129|0.127|0.129|0.181 | 0.147|0.145|0.144|0.204 |
| NCDE-FNO | 48769 | 1.734 | 0.031|0.040|0.054|0.070 | 0.032|0.041|0.050|0.070 |
| DeepONet | 4329472 | 0.009 | 0.109|0.118|0.123|0.174 | x |
| FNO | 4924449 | 0.166 | 0.019|0.019|0.019|0.027 | x |
| NSPDE | 3283457 | 0.156 | 0.001|0.001|0.003|0.004 | 0.001|0.001|0.003|0.004 |
| DLR-Net | 133178 | 0.110 | 0.003|0.001|0.002|0.001 | 0.002|0.001|0.002|0.001 |

## 4.1 MODELS' ROBUSTNESS TO DIFFERENT NOISE SIMULATIONS

We conducted experiments to evaluate the ML models' performance on datasets generated with different noise truncation degrees $J$. We use 1200 samples in total for each experiment, and each dataset is split into training, validation, and test sets with relative sizes $70\%/15\%/15\%$. We use the validation set for hyperparameter tuning and early stopping. For each model and each SPDE dataset with multiple noise truncation degrees, we search the hyperparameters for network training on dataset with one degree ($J = 128$) on the task "$\xi \to u$" and apply the best one in other cases.

The relative $L^2$-errors for several models on the dynamic $\Phi_1^4$ model are shown in Table 3. We also investigated the performance of the baseline models in the Dynamic $\Phi_1^4$ Model with $\sigma = 1$, a larger-scale parameter before the noise term. The results are reported in Table F3 in Appendix. We observe that DLR-Net, NSPDE and FNO achieve the top three highest accuracy, and the performance of all models (except DLR-Net) degrades at $J = 256$. Furthermore, we plotted the prediction curves for the three well-performed models DLR-Net, FNO and NSPDE in Figure G5. We observe that FNO and NSPDE show deviations from the ground truth (with $J = 128$), particularly in regions with larger predicted x-values and x-values near boundaries in mean prediction, while DLR-Net demonstrates strong agreement with the ground truth. Moreover, in variance prediction, the deviations of FNO and NSPDE from the ground truth variance are more pronounced than the mean predictions. All the results indicate that the regularity feature in DLR-Net help it robustly predict the ground truth, as measured by relative $L^2$ error, mean deviation and variance deviation across various noise degrees.

## 4.2 EVALUATION UNDER DIFFERENT METRICS

We compare different evaluation metrics of the baseline models on the test set of dynamic $\Phi_1^4$ model with $J = 256$ and $\sigma = 0.1$. All the models are trained with relative $L^2$-error as the training objective. The results are reported in Table 4. We observe that NSPDE and DLR-Net perform comparably, both achieving top-tier results for metrics relative $L^2$, $W^{1,2}$-Sobolev norm, auto-correlation and correlation. Notably, most models exhibit degraded performance under the $W^{1,2}$-Sobolev norm except DLR-Net. One potential reason is that $W^{1,2}$-Sobolev norm takes the derivatives of the spatio-temporal solution into account, and the calculation of the regularity feature in the DLR-Net regularize the derivatives, while other baselines do not regularize the higher-order derivatives.

## 4.3 COMPARISON ON SCALING LAW

In this section, we select four well-performed models in SPDE learning tasks, including NSPDE, DLR-Net, DeepONet and FNO as representatives to test the scaling law with respect to the sample size and network size on the dynamic $\Phi_1^4$ model. For each experimental setting, the dataset is divided into training/validation/test sets using a 4:1:1 ratio. We investigate the scaling laws of the models' relative $L^2$-error on test data with respect to model size and dataset size. From the results in Table 5, we have two main observations: first, the performance improve slightly as the amount of data increases for all the four models; second, FNO demonstrates relatively good scaling capability when the network

Table 4: Different test metrics of models on the test set of $\Phi_1^4$ with $J = 256$ and $\sigma = 0.1$. "Autocor" and "Cor" are abbreviations for the auto-correlation and correlation. Lower values are better.

| Model | $\xi \mapsto u$ | $(u_0, \xi) \mapsto u$ |
|---|---|---|
| | $L^2|W^{1,2}$-Sobolev$|$Autocor$|$Cor | $L^2|W^{1,2}$-Sobolev$|$Autocor$|$Cor |
| NCDE | 0.099|0.966|0.796|0.090 | 0.147|0.919|1.664|0.167 |
| NRDE | 0.181|0.864|1.056|0.257 | 0.204|0.865|1.287|0.261 |
| NCDE-FNO | 0.070|0.807|0.077|0.014 | 0.070|0.754|0.123|0.015 |
| DeepONet | 0.174|0.857|1.162|0.229 | x |
| FNO | 0.027|0.608|0.082|0.009 | x |
| NSPDE | 0.004|0.061|0.006|0.0006 | 0.004|0.060|0.003|0.0006 |
| DLR-Net | 0.001|0.005|0.001|0.0001 | 0.001|0.005|0.0003|0.00006 |

width expands, suggesting that wider network size is essential for FNO to adequately capture the required feature representations.

Table 5: Relative $L^2$ test error for task $\xi \to u$ on the dynamic $\Phi_1^4$ model with different training and evaluation sample size and network size. Data is generated with $J = 128$ and $\sigma = 1$. The sample size refers to the total amount of the training and validation sets. $nW$ (or $nD$) denotes $n$ times of the network width (or the network depth). "/" means inapplicable as the base model depth is $D = 1$.

| Model / Sample Size | 1000 | 2000 | 3000 | 5000 | 10000 |
|---|---|---|---|---|---|
| DeepONet | 0.840752 | 0.821926 | 0.794465 | 0.738755 | 0.600467 |
| FNO | 0.133919 | 0.133818 | 0.132809 | 0.131829 | 0.128889 |
| NSPDE | 0.018516 | 0.018482 | 0.018378 | 0.018193 | 0.018185 |
| DLR-Net | 0.000759 | 0.000722 | 0.000674 | 0.000579 | 0.000475 |
| Model / Network Size | W*D | (2W)*D | (4W)*D | W*(2D) | W*(D/2) |
| DeepONet | 0.840752 | 0.845179 | 0.842743 | 0.845750 | 0.861783 |
| FNO | 0.133919 | 0.066011 | 0.005871 | 0.137363 | / |
| NSPDE | 0.018516 | 0.018505 | 0.018502 | 0.018674 | / |
| DLR-Net | 0.000759 | 0.000698 | 0.000475 | 0.003232 | 0.000896 |

## 4.4 APPROXIMATION ERROR ON SINGULAR SPDE

We use dataset of the singular dynamic $\Phi_2^4$ model to study how noise truncation degree and renormalization on affect the performance of the ML-based models. Specifically, we train NSPDE and its variant NSPDE-S and evaluate their approximation error against the true solution. These datasets are generated both with and without renormalization, employing various noise truncation degrees (noise level in Table 6). Each dataset contains 1200 samples in total and each is split into training, validation, and test sets with relative sizes $70\%/15\%/15\%$. To distinguish the models trained under different datasets, we use the denotation NSPDE$_{re}$ (or NSPDE-S$_{re}$) and NSPDE$_{ex}$ to represent the NSPDE (or NSPDE-S) model trained on datasets with renormalization and without renormalization, respectively. We evaluate the trained models under two testing regimes: (i) "in-distribution testing", where the train and test data are generated using the same solver configuration, and (ii) "testing on $\mathcal{D}_{128}^{re}$", the test data generated with the highest noise level $J = 128$ and renormalization, which serves as a proxy of ground truth.

As shown in Table 6, NSPDE-S$_{re}$, which incorporates an an additional input channel for the renormalization parameter, achieves significantly lower approximation error when testing on $\mathcal{D}128^{re}$ and delivers more consistent in-distribution testing performance across different $J$. This demonstrates that providing the renormalization constant as input significantly improves the model performance.

We further conduct multi-task learning experiments by combining $5 * 1200$ samples generated with renormalization under $J = 2, 8, 32, 64, 128$ into a training dataset. Models trained on this dataset are denoted as NSPDE$_{re}^{mix}$ and NSPDE-S$_{re}^{mix}$. Their performance is then evaluated on $\mathcal{D}_{128}^{re}$. The results in Table 7 demonstrates that in the multi-task learning, the additional input channel for renormalization

Table 6: Relative $L^2$-error for task $(u_0, \xi) \to u$ on the test set of the dynamic $\Phi_2^4$ Model. Data are generated by implementing renormalization method. The number located in "line NSPDE$_{ex}$" and "column $J = 2$" means the error for NSPDE$_{ex}$ model trained on data generated with $J = 2$.

| Model/Noise Level | $J = 2$ | $J = 8$ | $J = 32$ | $J = 64$ | $J = 128$ |
|---|---|---|---|---|---|
| NSPDE$_{re}$ ("in-dis" testing) | 0.010 | 0.017 | 0.052 | 0.112 | 0.223 |
| NSPDE$_{re}$ (testing on $\mathcal{D}_{128}^{re}$) | 2.805 | 1.318 | 1.180 | 0.475 | 0.223 |
| NSPDE-S$_{re}$ ("in-dis" testing) | 0.042 | 0.014 | 0.028 | 0.023 | 0.029 |
| NSPDE-S$_{re}$ (testing on $\mathcal{D}_{128}^{re}$) | 0.998 | 0.977 | 0.947 | 0.755 | 0.029 |
| NSPDE$_{ex}$ ("in-dis" testing) | 0.010 | 0.016 | 0.070 | 0.138 | 0.260 |
| NSPDE$_{ex}$ (testing on $\mathcal{D}_{128}^{re}$) | 3.135 | 1.412 | 1.187 | 0.422 | 0.244 |

Table 7: Relative $L^2$-error for task $(u_0, \xi) \to u$ on the mixed test data set of the dynamic $\Phi_2^4$ model. Data are generated with renormalization.

| Model/Test dataset | #Para | Time (ms) | $\mathcal{D}_2^{re}$ | $\mathcal{D}_8^{re}$ | $\mathcal{D}_{32}^{re}$ | $\mathcal{D}_{64}^{re}$ | $\mathcal{D}_{128}^{re}$ |
|---|---|---|---|---|---|---|---|
| Solver (Reno) | x | 129.291 | x | x | x | x | x |
| Solver (Expl) | x | 103.770 | x | x | x | x | x |
| NSPDE$_{re}^{mix}$ | 2103809 | 0.471 | 0.003 | 0.015 | 0.059 | 0.120 | 0.228 |
| NSPDE-S$_{re}^{mix}$ | 2103809 | 0.479 | 0.005 | 0.074 | 0.194 | 0.274 | 0.249 |

parameter that needs to be fitted has made the learning process slightly more challenging which results in a higher testing error of NSPDE-S.

## 4.5 Inference Time Comparison

We compare the per-sample inference time of the numerical solvers with that of ML models. We denote the numerical simulation for the singular SPDE after renormalization as "Solver (Reno)" and that for the singular SPDE without renormalizaiton as "Solver (Expl)" in Table. 7. Timings for the two numerical solvers were recorded on an Intel Xeon Platinum 8352V CPU, whereas the ML models' inference was measured on a RTX 4090(24GB) GPU with PyTorch@2.4 and CUDA@12.1. Despite the difference in devices, these measurements offer insight into the relative speed of the numerical solvers and ML models. The inference time of $\Phi_1^4$ and $\Phi_2^4$ is reported in Table 3 and Table 7 respectively. When timing the numerical solvers, we use the same data resolutions as in the initial data generation (Table 1). When timing the ML models, the resolution of test data is the same as in model training. We measure the average time each ML model takes to perform inference on a batch (batch size = 100) of samples, and then divide it by the batch size to obtain the average inference time per sample. The results show that the ML models significantly improve simulation efficiency.

## 5 Discussions and Limitations

In this work, we introduce SPDEBench, an extensive benchmark for machine learning SPDEs, which includes singular and non-singular SPDE simulated datasets and baseline ML surrogate models. SPDEBench is among the very first ML implementations of singular SPDEs' solutions in the framework of regularity structures. It is expected to shed new light on ML methodologies for SPDEs, especially on the developments of novel designs for AI approaches to the singular SPDEs. While real-world datasets are not included in this study, the in-depth analyses of ML models in SPDEBench offer valuable insights for modeling real-world stochastic systems and for studying how predictions respond to spatio-temporal noise. To prevent misinterpretation or over-reliance, SPDEBench ensures transparency in data generation and evaluation, especially given the challenges posed by singularities and numerical artifacts. Our dataset currently emphasizes certain equations and ML models, potentially underrepresenting others that might be important in different domains. For instance, three-dimensional and higher-dimensional SPDEs, as well as the transformer-based ML models are not yet included due to computational resource constraints. These limitations will be addressed through community contributions and by extending SPDEBench to cover a broader range of equations, ML algorithms, and parameter regimes.

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

## A  BROADER IMPACT, ETHICS STATEMENT AND REPRODUCIBILITY

The proposed SPDEBench dataset and ML models will benefit researchers and practitioners in applied mathematics, computational physics, and engineering who rely on efficient surrogates for expensive or intractable numerical simulations. It also supports the machine learning community in developing and benchmarking models capable of capturing spatio-temporal roughness in data, thus facilitating broader advancements in scientific machine learning.

We are not aware of any potential ethical concerns arising from this research, as it does not involve human subjects, sensitive data, or foreseeable negative societal impacts. All datasets used are publicly available for academic purposes. To ensure the reproducibility of our results, we have provided the source code, detailed experimental configurations, and data preprocessing scripts in our supplementary materials.

## B  USE OF LLMS

We use Large Language Model, e.g., DeepSeek to aid and polish the English writing.

## C  BASIC CONCEPTS IN SPDE THEORY

### C.1  SPACE-TIME WHITE NOISE

In most cases, the noise term $\xi = \xi(t, x)$ is introduced to represent a so-called space-time white noise. Noise of this type can be viewed as a "delta-correlated" random distribution. Formally, a space-time white noise in $\mathbb{R}^n$ is a Gaussian random variable with mean zero and correlation:

$$\mathbb{E}\left[\xi(t,x)\xi(s,y)\right] = \delta(t-s)\delta(x-y),$$

where $\delta$ denotes the delta function. It, of course, is not an actual function but a distribution, and $\xi$ should be interpreted as a random distribution. That is, $\xi$ is a generalized centred Gaussian random field, with covariance defined in the sense of test functions:

$$\mathbb{E}\left[\langle \xi, h \rangle \langle \xi, k \rangle\right] = \int_0^T \int_{\mathbb{R}^n} h(t,x)k(t,x)\,\mathrm{d}x\mathrm{d}t, \quad \forall h, k \in L^2([0,T] \times \mathbf{R}^n).$$

This is equivalent to the time derivative of the cylindrical Wiener process defined in Section 2.2.

As a slight variant, noise term white in time and "colored" in space can be introduced as

$$\mathbb{E}\left[\xi(t,x)\xi(s,y)\right] = \delta(t-s)q(x-y),$$

where $q$ is the correlation function. The smoother $q$ is, the more regular is the noise. The colored noise terms can be represented as the time derivative of $Q$-Wiener processes in the next subsection.

## C.2 $Q$-WIENER PROCESS

Let $H$ be a separable Hilbert space with a complete orthonormal basis (ONB) $\{\phi_k\}_{k\in\mathbb{N}}$. Consider an operator $Q \in \mathcal{L}(H)$ such that there exists a bounded sequence of nonnegative real numbers $\{\lambda_k\}_{k\in\mathbb{N}}$ such that $Q\phi_k = \lambda_k\phi_k$ for all $k \in \mathbb{N}$ (this is implied by $Q$ being a trace class, non-negative, symmetric operator, for example).

**Definition C.1** ($Q$-Wiener process). Let $(\Omega, \mathcal{F}, \mathcal{F}_t, \mathbb{P})$ be a filtered probability space. A $H$-valued stochastic process $\{W(t) : t \geq 0\}$ is a $Q$-Wiener process if

1. $W(0) = 0$ almost surely;

2. $W(t; \omega)$ is a continuous sample trajectory $\mathbb{R}^+ \mapsto H$, for each $\omega \in \Omega$;

3. $W(t)$ is $\mathcal{F}_t$-adapted and $W(t) - W(s)$ is independent of $\mathcal{F}_s$ for $s < t$;

4. $W(t) - W(s) \sim \mathcal{N}(0, (t-s)Q)$ for all $0 \leq s \leq t$.

In analogy to the Karhunen–Loève expansion, it can be shown that $W(t)$ is a $Q$-Wiener process if and only if for all $t \geq 0$,

$$W(t) = \sum_{j=1}^{\infty} \sqrt{\lambda_j}\phi_j\beta_j(t),$$

where $\beta_j(t)$ are i.i.d. Brownian motions, and the series converges in $L^2(\Omega, H)$. Moreover, the series is $\mathbb{P}$-a.s. uniformly convergent on $[0,T]$ for arbitrary $T > 0$ (i.e., converges in $L^2(\Omega, C([0,T], H))$). One should note that formally if we let $Q = I$, $Q$-Wiener process becomes cylindrical Wiener process which converges in a larger space.

In dimension 1, we set the domain $D = (0, L)$ and consider an $H^r_{per}(D)$-valued $Q$-Wiener process $W(t)$, for a given $r \geq 0$. (In the KdV case, we take $r = 2$.) We take the ONB $\phi_k(x) = \sqrt{2/L}\sin(k\pi x/L)$ with corresponding $\lambda_j = (\lfloor j/2 \rfloor + 1)^{-(2r+1+\epsilon)}$ for an $\epsilon > 0$ (we take $\epsilon = 0.001$), and sample from the truncated expansion $W^J(t) = \sum_{j=1}^J \phi_j\beta_j(t)$ at the points $x_k = kL/N$ for $k = 1, \ldots, N$.

In dimension 2, we set $D = (0, L_x) \times (0, L_y)$, take the $L^2(D)$-ONB $\phi_{j,k}(x,y) = 1/\sqrt{L_x L_y}e^{2i\pi(jx/L_1 + ky/L_2)}$ with corresponding $\lambda_{j,k} = e^{-\alpha(j^2+k^2)}$, for a parameter $\alpha > 0$. (In the NSE case, we take $\alpha = 0.005$.) Then we sample from the truncations $W^{J_x, J_y}(t) = \sum_{j=-J_x/2+1}^{J_x/2} \sum_{k=-J_y/2+1}^{J_y/2} \sqrt{\lambda_{j,k}}\phi_{j,k}\beta_{j,k}(t)$ at $x_m = mL_x/N_x$, $y_n = nL_y/N_y$ for $m = 1, \ldots, N_x; n = 1, \ldots, N_y$.

## C.3 THE ILL-POSEDNESS OF THE DYNAMIC $\Phi_2^4$ MODEL

We begin with the definition of Hölder–Besov space. Let $\chi, \theta$ be nonnegative radial functions on $\mathbb{R}^d$, such that

i. $\mathrm{supp}(\chi)$, the support of $\chi$, is contained in a ball and the support of $\theta$ is contained in an annulus;

ii. $\chi(z) + \sum_{j \geqslant 0} \theta(2^{-j}z) = 1$ for all $z \in \mathbb{R}^d$;

iii. $\mathrm{supp}(\chi) \cap \mathrm{supp}(\theta(2^{-j}\cdot)) = \emptyset$ for $j \geqslant 1$ and $\mathrm{supp}(\theta(2^{-i}\cdot)) \cap \mathrm{supp}(\theta(2^{-j}\cdot)) = \emptyset$ for $|i - j| > 1$.

We call such $(\chi, \theta)$ a dyadic partition of unity, for whose existence we refer to (Bahouri et al., 2011, Proposition 2.10). Let $\mathcal{F}$ be the Fourier operator, the Littlewood–Paley blocks are now defined as

$$\Delta_{-1}u = \mathcal{F}^{-1}(\chi \mathcal{F}u) \quad \Delta_j u = \mathcal{F}^{-1}(\theta(2^{-j}\cdot)\mathcal{F}u).$$

For $\alpha \in \mathbb{R}$, $p, q \in [1, \infty]$, we define

$$\|u\|_{B_{p,q}^{\alpha}} := \big( \sum_{j \geqslant -1} (2^{j\alpha}\|\Delta_j u\|_{L^p})^q \big)^{1/q},$$

with the usual interpretation as $L^{\infty}$ norm in case $q = \infty$. The Besov space $B_{p,q}^{\alpha}$ consists of the completion of smooth functions with respect to this norm and the Hölder–Besov space $\mathcal{C}^{\alpha}$ is given by $\mathcal{C}^{\alpha} = B_{\infty,\infty}^{\alpha}$. We point out that everything above can be applied to distributions on the torus considering periodic spaces. For $\alpha > 0$, $\mathcal{C}^{\alpha}$ is equivalent to Hölder space. See Triebel (2006) for details.

In the case of dynamic $\Phi_2^4$ model, the space-time white noise $\xi \in \mathcal{C}^{-2-\alpha}(\forall \alpha > 0)$Hairer (2014). The equation now falls in the subcritical regime in the language of Hairer's regularity structures theory. This means the regularity of solution $u$ is the same as the regularity of the stochastic convolution $X$, the model can be understood as a perturbation of the solution to the linear equation. By standard Schauder's estimates in the elliptic PDE theory, $u$ and $X$ are expected to take values in space $\mathcal{C}^{-\alpha}(\forall \alpha > 0)$, which means $u$ is not a function but a distribution and leaves the non-linear term $u^3$ undefined in the classical sense. See Hairer (2014); Gubinelli & Hofmanová (2019) for details.

If we use Galerkin projection $W^J$ for the noise and $a_{J^{-1}}$ as approxiamtion for the renormalization with truncation degree $J$(see section 3.1.2), denotes $u^J$ the approximation solution, then the following convergence rate holds (see Ma & Zhu (2021)): let $\alpha \in (0, 2/9)$, $\gamma' > 3\alpha/2$, for any $\delta > 0$, there exists constant $C$ and $\delta$ with arbitrary small positive value such that

$$\left( \mathbb{E} \sup_{t \in [0,T]} t^{2\gamma'} \|u(t) - u^J(t)\|_{\mathcal{C}^{-\alpha}}^2 \right)^{\frac{1}{2}} \leq \frac{C}{J^{\alpha - \delta}}.$$

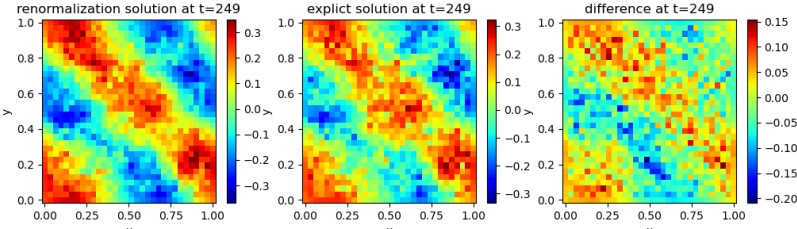

Figure C1: Comparison of data generated of $\Phi_2^4$ when truncation degree $J = 128$ at time step $t = 249$. **Left panel:** Mean of 1200 solution samples generated by numerically solve Eqn (4) under 1200 sampled $(u_0, \xi)$ pairs, while the **Middle Panel** is mean of 1200 solution samples generated by numerically solve Eqn (3) without implementing the renormalization.

## C.4 NONSINGULAR SPDE DATASETS

The nonsingular SPDE datasets include 1D Dynamic $\Phi_1^4$ Model, 1D Korteweg–De Vries (KDV) equation, 1D wave equation and the 2D incompressible Navier–Stokes equation (NSE, in the form of vorticity equation). For each equation with one noise truncation degree, we generate 10000 samples. In Table 1 and Table 2, we list the main hyper-parameters and configurations in the numerical solvers for generating these datasets. The numerical simulation methods employed in this section reference the literature by Salvi et al. (2022); Chevyrev et al. (2024).

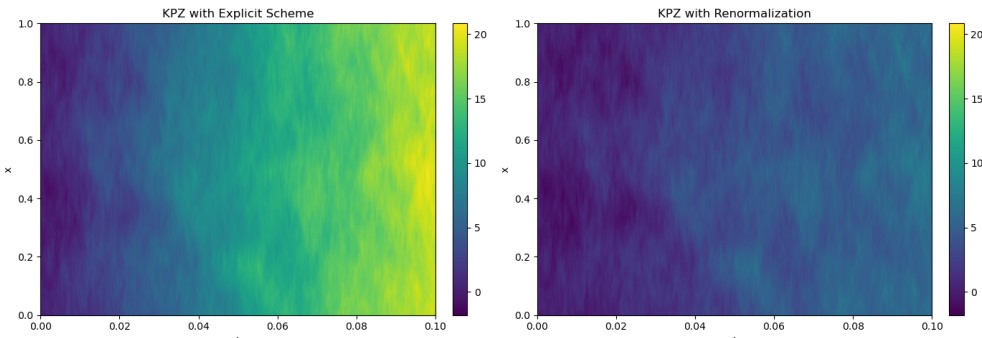

Figure C2: Comparison of data generated of KPZ when truncation degree $J = 256$ at time interval $[0, 0.1]$. The right panel is a solution generated by numerically solve Eqn (6), while the left panel is a solution generated by numerically solve Eqn (5) without implementing the renormalization.

**Dynamic $\Phi_1^4$ Model** is a basic model in superconductivity and quantum field theory that essentially demonstrates parabolic SPDE phenomena, and the stochastic $\Phi_d^4$ model is a typical example in the theory of regularity structure Lin (1996); Hairer (2014). Consider the Dynamic $\Phi_1^4$ Model:

$$\begin{cases} \partial_t u - \Delta u = 3u - u^3 + \sigma\xi & \text{in } [0, 0.05] \times \mathbf{T}^1, \\ u\big|_{t=0} = u_0 & \text{at } \{0\} \times \mathbf{T}^1. \end{cases}$$

Following the setup in previous work Salvi et al. (2022), we set the initial condition as $u_0(x) = x(1-x) + \kappa\eta(x)$ with $\eta(x) = \sum_{k=-10}^{k=10} \frac{a_k}{(|k|+1)^2} \sin(2k\pi x)$, where $a_k \overset{\text{i.i.d.}}{\sim} \mathcal{N}(0,1)$. We take $\kappa = 0$ or $\kappa = 0.1$ to generate datasets with fixed or varying initial conditions, respectively. We use finite difference method to solve the SPDE and the response paths are generated using 128 evenly distanced points in space and $\Delta t = 10^{-3}$ in time. The space-time white noise is sampled from the truncated cylindrical Wiener process and scaled by $\sigma = 1$

**Korteweg–De Vries (KdV) Equation** is a typical dispersive SPDE arising from the study of water waves, and it plays an important role in the theory of solitons and integrable systems Killip & Vişan (2019). Consider the KdV Equation for shallow water:

$$\begin{cases} \partial_t u - 0.001\partial_{xx} u + \gamma\partial_{xxx} u = 6u\partial_x u + \sigma\xi & \text{in } [0, 0.5] \times \mathbf{T}^1, \\ u\big|_{t=0} = u_0 & \text{at } \{0\} \times \mathbf{T}^1. \end{cases}$$

In our experiments, we take $\gamma = 0.1$. The initial datum is $u_0(x) = \sin(2\pi x) + \kappa\eta(x)$, where $\eta$ is defined as for the dynamic $\Phi_1^4$ model. Similarly, we take $\kappa = 0$ or $\kappa = 0.1$ to generate datasets with fixed or varying initial conditions, respectively. The numerical solution was calculated using the spectral Galerkin method. The space-time white noise is sampled from the truncated cylindrical Wiener process and scaled by $\sigma = 0.5$. The detailed data simulation approach for KdV equation including the time step $\Delta t$ follows previous work Salvi et al. (2022).

**Wave Equation** is of the hyperbolic type, featuring finite speed of propagation and energy conservation, which is closely related to applications in geophysics Kenig & Merle (2008); Gubinelli et al. (2023). Consider the wave Equation:

$$\begin{cases} \partial_{tt} u - \partial_{xx} u = \cos(\pi u) + u^2 + u \cdot \xi & \text{in } [0, 0.5] \times \mathbf{T}^1, \\ (u, \partial_t u)\big|_{t=0}(x) = \big(u_0(x), v_0(x)\big) & \text{at } \{0\} \times \mathbf{T}^1. \end{cases}$$

We set $u_0(x) = \sin(2\pi x) + \kappa\eta(x)$ and $v_0(x) = x(1-x)$, where $\eta(x)$ is as before and $\kappa$ is set to be 0 or 0.1. The numerical solution was calculated with both temporally and spatially 2nd-order central difference scheme. The space-time white noise is sampled from truncated cylindrical Wiener process. In data simulation process, we set a time step size $\Delta t = 10^{-3}$ Chevyrev et al. (2024).

**Incompressible Navier–Stokes equation** is the fundamental equation in mathematical hydrodynamics, whose well-posedness theory is at the heart of the mathematical analysis of SPDEs Flandoli &

Gatarek (1995); Temam (2024). We consider the (scalar) *vorticity equation* on the 2D torus, obtained by taking the curl of the Navier–Stokes Equation:

$$\begin{cases} \partial_t \omega - \nu \Delta \omega = -u \cdot \nabla \omega + f + \sigma \xi & \text{in } [0,1] \times \mathbf{T}^2, \\ \omega\big|_{t=0} = \omega_0 & \text{at } \{0\} \times \mathbf{T}^2. \end{cases}$$

Here, the velocity field $u = [u^1, u^2]^\top$ is incompressible ($\text{div}\, u = 0$), and the vorticity is defined as $\omega = \text{curl}\, u = [\partial_2 u^1, -\partial_1 u^2]^\top$. In experiments, we take the coefficients $\nu = 10^{-4}$, $\sigma = 0.005$, and $f = 0.1(\sin(2\pi(x+y)) + \cos(2\pi(x+y)))$ Li et al. (2020). The initial condition is generated according to $\omega_0 \sim \mu$, where $\mu = \mathcal{N}(0, 3^{3/2}(-\Delta + 9I)^{-3})$. The numerical solution was calculated using the spectral Galerkin method. The colored-in-space noise is the sum of 10 trajectories sampled from truncated Q-Wiener process. The detailed data simulation approach for KdV equation including the time step $\Delta t$ and spatial downscaling method follows previous work Salvi et al. (2022).

## D  ADDITIONAL DEMO FOR USERS

```
from SPDE_HACKATHON.model.NSPDE.utilities import *
parquetfile = ".../Phi42+_expl_xi_eps_2_1200.parquet"
data_path = ".../Phi42"
get_data_Phi42(parquetfile, data_path)
data = scipy.io.loadmat(".../Phi42mat_data.mat")
W, Sol = data['W'], data['sol']
xi = torch.from_numpy(W.astype(np.float32))
data = torch.from_numpy(Sol.astype(np.float32))
train_loader, test_loader = dataloader_nspde_2d(data, xi, ntrain, ntest, T,
sub_t, sub_x, batch_size)
```

Listing 1: Using the Pytorch data loader

The datasets are stored in PARQUET format. File naming format is {SPDE name}-{Tasks}-{Truncation degree}-{Sample size}.parquet. For the $\Phi_1^4$ equation, different value of the parameter $\sigma$ are denoted by *01* and *1*. For the $\Phi_2^4$ equation, data generated with renormalization and without renormalization are denoted by *reno* and *expl*.

## E  EXPERIMENTAL DETAILS

For all experiments, the dataset is split into training, validation and test sets with relative sizes $70\%/15\%/15\%$. We use the validation set for hyperparameter tuning and early stopping. For each model and each SPDE dataset with multiple noise truncation degrees, we search the hyperparameters for network training on dataset with one degree ($J = 128$) on the task "$\xi \to u$" and apply the best one in other cases. For all the baseline models except DLR-Net, we follow the grid search schemes for hyperparameter selection in Salvi et al. (2022); for the DLR-Net, since the numerical-solution setups of our $\Phi_1^4$ and NSE datasets are similar to what has been experimented in Gong et al. (2023), we directly employ their configuration. More details about our experiments (such as the optimizer, scheduler, and the specific values of hyperparameters we used) can be found in our Supplementary for codes. We report part of the grid search results in Table E1 and Table E2.

## F  ADDITIONAL EXPERIMENTAL RESULTS

### F.1  RELATIVE $L^2$-ERROR AND STATISTICAL SIGNIFICANCE ON MORE DATASETS

In this section, we show more experimental results in Table F4 to F8. As the performance of FNO, NSPDE, DLR-Net is significantly better than other models, we mainly evaluate the performance of these three models on KdV, wave equation and incompressible NSE. For DLR-Net, it requires to develop the regularity feature layer for different SPDEs which is beyond the scope of this paper. Therefore, we only apply it on incompressible NSE according to its original work Gong et al.

Table E1: Grid search NSPDE ($\Phi_2^4$: $\mathcal{D}_{128}^{\text{re}}$)

| $d_h$ | Picard's iterations | modes 1 | modes 2 | modes 3 | # Para | validation loss |
|---|---|---|---|---|---|---|
| 32 | 1 | 8 | 8 | 8 | 530945 | 0.249 |
| 32 | 2 | 8 | 8 | 8 | 530945 | 0.249 |
| 32 | 3 | 8 | 8 | 8 | 530945 | 0.249 |
| 32 | 4 | 8 | 8 | 8 | 530945 | 0.251 |
| 32 | 1 | 8 | 16 | 8 | 1055233 | 0.238 |
| 32 | 2 | 8 | 16 | 8 | 1055233 | 0.239 |
| 32 | 3 | 8 | 16 | 8 | 1055233 | 0.239 |
| 32 | 4 | 8 | 16 | 8 | 1055233 | 0.239 |
| 32 | 1 | 16 | 8 | 8 | 1055233 | 0.238 |
| 32 | 2 | 16 | 8 | 8 | 1055233 | 0.235 |
| 32 | 3 | 16 | 8 | 8 | 1055233 | 0.237 |
| 32 | 4 | 16 | 8 | 8 | 1055233 | 0.238 |
| 32 | 1 | 16 | 16 | 8 | 2103809 | 0.239 |
| 32 | 2 | 16 | 16 | 8 | 2103809 | 0.238 |
| 32 | 3 | 16 | 16 | 8 | 2103809 | 0.239 |
| 32 | 4 | 16 | 16 | 8 | 2103809 | 0.238 |

Table E2: Grid search NSPDE-S ($\Phi_2^4$: $\mathcal{D}_{128}^{\text{re}}$)

| $d_h$ | Picard's iterations | modes 1 | modes 2 | modes 3 | # Para | validation loss |
|---|---|---|---|---|---|---|
| 32 | 1 | 8 | 8 | 8 | 530945 | 0.030 |
| 32 | 2 | 8 | 8 | 8 | 530945 | 0.024 |
| 32 | 3 | 8 | 8 | 8 | 530945 | 0.028 |
| 32 | 4 | 8 | 8 | 8 | 530945 | 0.028 |
| 32 | 1 | 8 | 16 | 8 | 1055233 | 0.022 |
| 32 | 2 | 8 | 16 | 8 | 1055233 | 0.021 |
| 32 | 3 | 8 | 16 | 8 | 1055233 | 0.018 |
| 32 | 4 | 8 | 16 | 8 | 1055233 | 0.025 |
| 32 | 1 | 16 | 8 | 8 | 1055233 | 0.020 |
| 32 | 2 | 16 | 8 | 8 | 1055233 | 0.021 |
| 32 | 3 | 16 | 8 | 8 | 1055233 | 0.025 |
| 32 | 4 | 16 | 8 | 8 | 1055233 | 0.025 |
| 32 | 1 | 16 | 16 | 8 | 2103809 | 0.014 |
| 32 | 2 | 16 | 16 | 8 | 2103809 | 0.016 |
| 32 | 3 | 16 | 16 | 8 | 2103809 | 0.021 |
| 32 | 4 | 16 | 16 | 8 | 2103809 | 0.028 |

(2023). For the incompressible NSE, in the task $\xi \mapsto u$, the initial condition is sampled from $\mathcal{N}(0, 3^2(-\Delta + 9I)^{-3})$ and fixed. In the task $(u_0, \xi) \mapsto u$, the initial condition is $w^\star + w_0$, where $w^\star$ is sampled from $\mathcal{N}(0, 3^2(-\Delta + 9I)^{-3})$ and fixed and $w_0 \sim \mathcal{N}(0, 3^2(-\Delta + 9I)^{-3})$. Figure F3 illustrates the prediction results from one of the experiments on NSE. Moreover, we generate 500 additional samples in the case of NSE($J = 256$) and dynamics $\Phi_2^4(J = 128)$ and test the pre-trained models on them to evaluate the model loss error. Results are reported in Table F3 and Table F14.

Table F4: Relative $L^2$-error on the test set of KdV. Data is generated with different truncation degrees ($J = 32|64|128|256$). ($\xi$ is sampled from a cylindrical Wiener process and $\sigma = 0.5$)

| Model | $\xi \mapsto u$ $J = 32|64|128|256$ | $(u_0, \xi) \mapsto u$ $J = 32|64|128|256$ |
|---|---|---|
| FNO | 0.072|0.071|0.118|0.166 | x |
| NSPDE | 0.010|0.012|0.093|0.134 | 0.013|0.015|0.090|0.126 |

Table F3: Relative $L^2$-error on the test set of the dynamic $\Phi_1^4$ model. Data is generated with different truncation degree ($J = 32|64|128|256$) and $\sigma = 1$.

| Model | $\xi \mapsto u$ $J = 32\|64\|128\|256$ | $(u_0, \xi) \mapsto u$ $J = 32\|64\|128\|256$ |
|---|---|---|
| NCDE | 0.509\|0.497\|0.510\|0.632 | 0.506\|0.546\|0.593\|0.633 |
| NRDE | 0.868\|0.822\|0.856\|0.944 | 0.846\|0.855\|0.863\|0.935 |
| NCDE-FNO | 0.215\|0.224\|0.270\|0.299 | 0.196\|0.226\|0.311\|0.308 |
| DeepONet | 0.783\|0.784\|0.841\|0.923 | x |
| FNO | 0.132\|0.126\|0.134\|0.148 | x |
| NSPDE | 0.009\|0.011\|0.019\|0.021 | 0.010\|0.009\|0.021\|0.022 |
| DLR-Net | 0.002\|0.001\|0.001\|0.001 | 0.001\|0.001\|0.001\|0.001 |

Table F5: Relative $L^2$-error on the test set of the KdV Equation. Data is generated with different truncation degrees ($J = 32|64|128|256$). ($\xi$ is sampled from a $Q$-Wiener process and $\sigma = 1$)

| Model | $\xi \mapsto u$ $J = 32\|64\|128\|256$ | $(u_0, \xi) \mapsto u$ $J = 32\|64\|128\|256$ |
|---|---|---|
| FNO | 0.045\|0.047\|0.046\|0.048 | x |
| NSPDE | 0.006\|0.006\|0.006\|0.006 | 0.009\|0.009\|0.008\|0.008 |

Table F6: Relative $L^2$-error on the test set of the wave Equation. Data is generated with different truncation degrees ($J = 32|64|128|256$).

| Model | $\xi \mapsto u$ $J = 32\|64\|128\|256$ | $(u_0, \xi) \mapsto u$ $J = 32\|64\|128\|256$ |
|---|---|---|
| FNO | 0.009\|0.011\|0.016\|0.026 | x |
| NSPDE | 0.007\|0.008\|0.014\|0.021 | 0.006\|0.007\|0.018\|0.024 |

Table F7: Relative $L^2$-error on the test set of the Navier–Stokes (vorticity) Equation. Data is generated with different truncation degrees ($J = 32|64|128|256$). The resolution for training and test data are $16 \times 16$.

| Model | $\xi \mapsto u$ $J = 32\|64\|128\|256$ | $(u_0, \xi) \mapsto u$ $J = 32\|64\|128\|256$ |
|---|---|---|
| FNO | 0.092\|0.090\|0.090\|0.090 | x |
| NSPDE | 0.037\|0.038\|0.037\|0.037 | 0.072\|0.053\|0.066\|0.063 |
| DLR | 0.023\|0.023\|0.023\|0.022 | 0.022\|0.033\|0.021\|0.026 |

Table F8: Statistics (mean ± standard deviation) of pre-trained models' prediction loss evaluated on an additionally generated NSE($J = 256$) test set of 500 samples

| Model | $\xi \mapsto u$ | $(u_0, \xi) \mapsto u$ |
|---|---|---|
| FNO | $0.090 \pm 0.007$ | x |
| NSPDE | $0.038 \pm 0.003$ | $0.061 \pm 0.026$ |
| DLR | $0.023 \pm 0.002$ | $0.026 \pm 0.007$ |

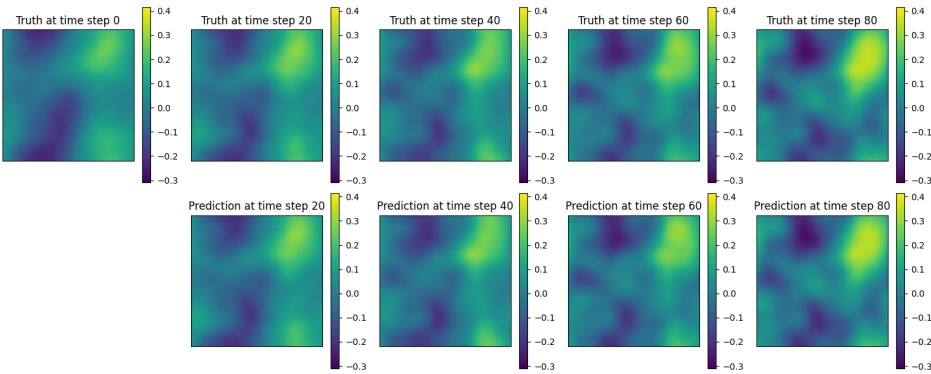

Figure F3: Illustration of the time evolution of the NSE($J =256$) data and NSPDE's predictions. The model is trained on a $16 \times 16$ mesh and evaluated on a $64 \times 64$ mesh for visualization. **Top panel**: Ground truth generated by the numerical solver. **Bottom panel**: Predictions from the model with input $(u_0, \xi)$.

Table F9: Statistics (mean ± standard deviation) of pre-trained model's prediction loss evaluated on an additionally generated test set of 500 samples for dynamic $\Phi_2^4$ model. Denote by $\mathcal{D}_J^{\text{re}}$ and $\mathcal{D}_J^{\text{ex}}$ the datasets constructed with and without renormalization, respectively. $\mathcal{D}_{128}^{\text{re}} \to \mathcal{D}_{128}^{\text{re}}$ means training on $\mathcal{D}_{128}^{\text{re}}$ and testing on $\mathcal{D}_{128}^{\text{re}}$.

| Tasks | mean + std |
|---|---|
| NSPDE ($\mathcal{D}_{128}^{\text{re}} \to \mathcal{D}_{128}^{\text{re}}, \xi \mapsto u$) | $0.240646 \pm 0.007078$ |
| NSPDE ($\mathcal{D}_{128}^{\text{re}} \to \mathcal{D}_{128}^{\text{re}}, (\xi, u_0) \mapsto u$) | $0.241992 \pm 0.002297$ |
| NSPDE ($\mathcal{D}_{128}^{\text{ex}} \to \mathcal{D}_{128}^{\text{re}}, \xi \mapsto u$) | $0.250163 \pm 0.002323$ |
| NSPDE ($\mathcal{D}_{128}^{\text{ex}} \to \mathcal{D}_{128}^{\text{re}}, (\xi, u_0) \mapsto u$) | $0.241992 \pm 0.002297$ |
| NSPDE-S ($\mathcal{D}_{128}^{\text{re}} \to \mathcal{D}_{128}^{\text{re}}, (\xi, a_\epsilon) \mapsto u$) | $0.029331 \pm 0.000064$ |
| NSPDE-S ($\mathcal{D}_{128}^{\text{re}} \to \mathcal{D}_{128}^{\text{re}}, (\xi, u_0, a_\epsilon) \mapsto u$) | $0.029308 \pm 0.000064$ |
| NSPDE-S ($\mathcal{D}^{\text{re}} \to \mathcal{D}_{128}^{\text{re}}, (\xi, a_\epsilon) \mapsto u$) | $0.295197 \pm 0.007920$ |
| NSPDE-S ($\mathcal{D}^{\text{re}} \to \mathcal{D}_{128}^{\text{re}}, (\xi, u_0, a_\epsilon) \mapsto u$) | $0.292941 \pm 0.008584$ |

Table F10: Ginzburg-Landau ($\sigma = 0.1, J = 256$), $\xi \mapsto u$, mean and std on a test set of 500 samples

| Model | mean + std |
|---|---|
| NCDE | $0.098 \pm 0.017$ |
| NRDE | $0.183 \pm 0.062$ |
| NCDE-FNO | $0.070 \pm 0.011$ |
| DeepONet | $0.175 \pm 0.058$ |
| FNO | $0.027 \pm 0.003$ |
| NSPDE | $0.0040 \pm 0.0006$ |
| DLR | $0.0005 \pm 0.0002$ |

Table F11: Ginzburg-Landau ($\sigma = 1, J = 256$), $\xi \mapsto u$, mean and std on a test set of 500 samples

| Model | mean + std |
|---|---|
| NCDE | $0.622 \pm 0.175$ |
| NRDE | $0.928 \pm 0.190$ |
| NCDE-FNO | $0.292 \pm 0.079$ |
| DeepONet | $0.915 \pm 0.188$ |
| FNO | $0.145 \pm 0.039$ |
| NSPDE | $0.021 \pm 0.006$ |
| DLR | $0.0011 \pm 0.0017$ |

Table F12: KdV ($\xi$ is sampled from a cylindrical Wiener process and $\sigma = 0.5, J = 256$), $\xi \mapsto u$, mean and std on a test set of 500 samples

| Model | mean + std |
|---|---|
| FNO | 0.165±0.041 |
| NSPDE | 0.132±0.041 |

Table F13: KdV ($\xi$ is sampled from a $Q$-Wiener process and $\sigma = 1, J = 256$), $\xi \mapsto u$, mean and std on a test set of 500 samples

| Model | mean + std |
|---|---|
| FNO | 0.047±0.012 |
| NSPDE | 0.006±0.001 |

Table F14: wave equation ($J = 256$), $\xi \mapsto u$, mean and std on a test set of 500 samples

| Model | mean + std |
|---|---|
| FNO | 0.026±0.004 |
| NSPDE | 0.021±0.002 |

## F.2 THE EFFECT OF NOISE APPROXIMATION METHOD

In this section, we explore the impact of the noise approximation method on model performance. On the one hand, we test the performance of FNO and NSPDE on two KdV datasets. On the other hand, we test the performance of WNO and FNO on two $\Phi_1^4$ datasets.

First, for the KdV datasets with noise generated via truncated cylindrical Wiener and a Q-Wiener processes, the results are shown in Figure F4, which compares the relative $L^2$-errors for FNO and NSPDE. Our results show that the performance of both FNO and NSPDE is sensitive to the truncation order of the cylindrical Wiener noise, with higher orders leading to worse prediction accuracy.

| Model | $\xi \mapsto u$ $J = 32\|64\|128\|256$ |
|---|---|
| FNO (Q-Wiener) | 0.045\|0.047\|0.046\|0.048 |
| NSPDE (Q-Wiener) | 0.006\|0.006\|0.006\|0.006 |
| FNO (cylindrical) | 0.072\|0.071\|0.118\|0.166 |
| NSPDE (cylindrical) | 0.010\|0.012\|0.093\|0.134 |

Figure F4: Comparison of FNO and NSPDE trained on KdV datasets with noise generated by cylindrical Wiener process ($\sigma = 0.5$) and Q-Wiener process ($\sigma = 1$).

Second, we test the hypothesis that the physics priors encoded in the network architecture exhibit an inductive bias toward the noise representation basis by comparing the wavelet and Fourier neural operators on Dynamic $\Phi_1^4$-F and Dynamic $\Phi_1^4$-H datasets. The results are shown in Table F15. We observe that the FNO model significantly and consistently outperforms the WNO model. For the Dynamic $\Phi_1^4$-F dataset, the FNO error (0.134 / 0.148) is approximately 70% lower than the WNO error (0.406 / 0.454). For the Dynamic $\Phi_1^4$-H dataset, the FNO error (0.139 / 0.175) is approximately 55-62% lower than the WNO error (0.306 / 0.458). Both models exhibit a slight increase in error when the resolution increases from $J = 128$ to $J = 256$. However, FNO demonstrates a smaller performance degradation, indicating better numerical stability.

Table F15: Models' relative $L^2$-error on test set for task $\xi \to u$ on the dynamic $\Phi_1^4$ with Fourier basis or Haar wavelet basis. Data is generated with $J = 128$ or $256$ and $\sigma = 1$.

| Model | Dynamic $\Phi_1^4$–F $J = 128\|256$ | Dynamic $\Phi_1^4$–H $J = 128\|256$ |
|---|---|---|
| FNO | 0.134\|0.148 | 0.139\|0.175 |
| WNO | 0.406\|0.454 | 0.306\|0.458 |

## G  COMPARISON BETWEEN PREDICTIONS AND GROUND TRUTH ACROSS MODELS

We benchmarked the performance of three models—DLR-Net, NSPDE, and FNO—on the task of predicting the solution of the dynamics $\Phi_1^4$ with truncation degree $J = 32$ and $J = 128$. The predicted solutions from each model were compared against the ground truth at both an intermediate time step (step 25) and the final time step (step 50). This comparison assessed the discrepancies in their mean predictions as well as the respective mean variances. The results are presented in Figure G5.

We have the following observations. First, the variance of the groundtruth with noise truncation degree $J = 128$ is significantly larger than that with noise truncation degree $J = 32$, showing the impact of the noise truncation degree on the statistics of the solution. Second, FNO and NSPDE show derivations from the groundtruth, particularly in regions with larger predicted x-values and x-values near boundaries in mean prediction (truncatin degree $J = 128$). Third, in variance prediction, the derivations of FNO and NSPDE from the groundtruth variance are even more pronounced than the mean predictions. All these results indicate that the regularity feature in DLR-Net make it robustly predict both the mean and variance across various noise degrees.

## H  SUPPLEMENT RESULTS ON EVALUATION UNDER DIFFERENT METRICS

Table H16: Test loss and training time of $\text{NSPDE}_{re}$ with $J = 128$ of dynamic $\Phi_2^4$ model with different training and testing loss.

| Loss | Relative $L^2$ Error | | | | $W^{1,2}$-Sobolev norm | | | |
|---|---|---|---|---|---|---|---|---|
| Epoch | 10 | 20 | 30 | 1000 | 10 | 20 | 30 | 1000 |
| Test loss | 0.4018 | 0.3511 | 0.3100 | 0.2407 | 0.5507 | 0.4123 | 0.3741 | 0.2184 |
| training time | 188.33 s/epoch | | | | 385.08 s/epoch | | | |

We test the evaluation metrics: relative $L^2$-error and $W^{1,2}$-Sobolev norm of the pre-trained models in the following table.

## I  DETAILS OF NSPDE-S

We incorporate the normalized $a_\epsilon$ as a scaling factor by multiplying it with the latent space variables in the NSPDE model, the implementation code is

```
a_eps = a_eps.unsqueeze(1)
zs = self.solver(z0,xi,grid)

gate = torch.sigmoid(torch.abs(a_eps)) * 2
zs = zs * gate
```

Listing 2: Renormalization Factor in NSPDE-S

The renormalization constant $a_\epsilon$ is normalized as a gate.

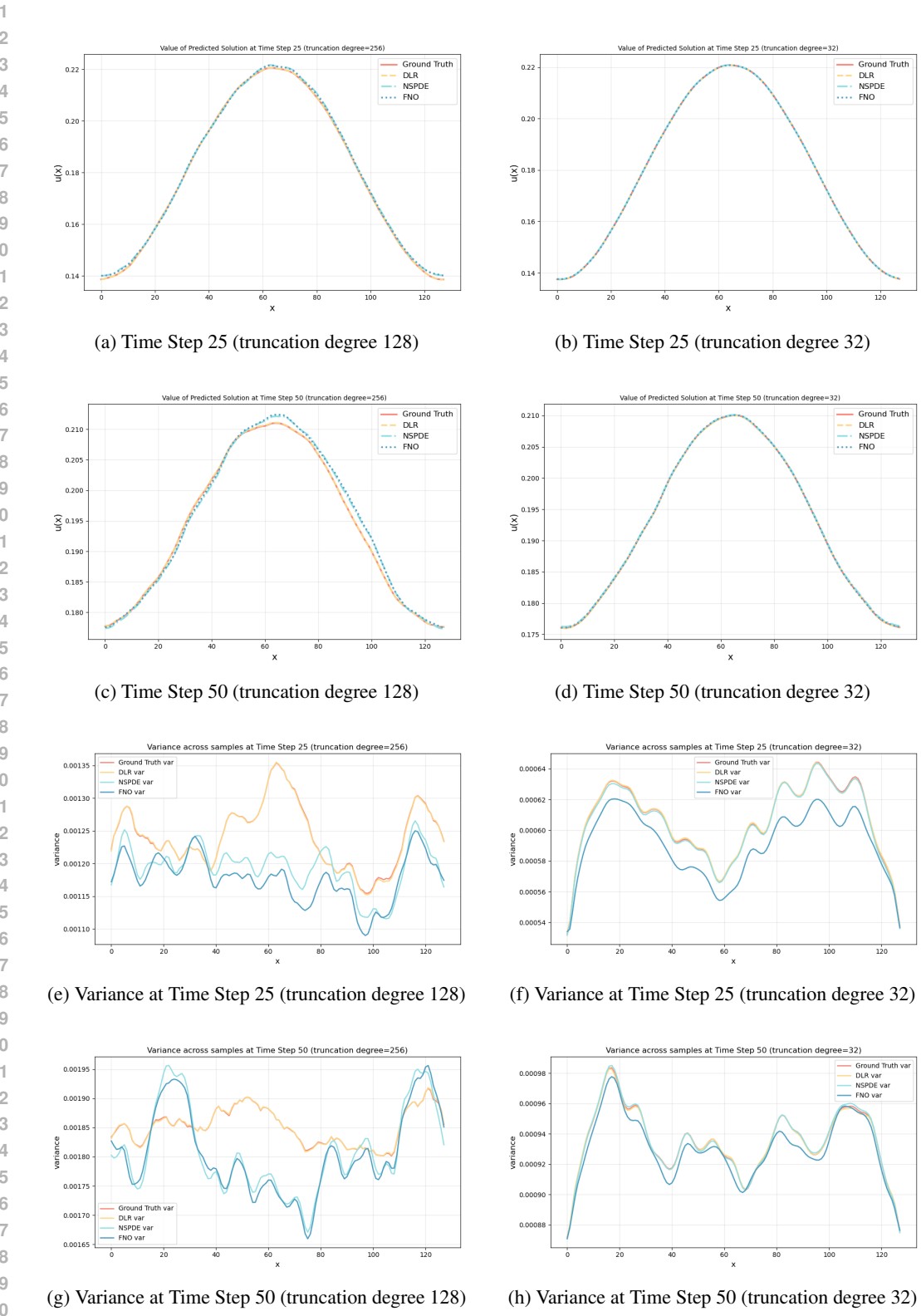

(a) Time Step 25 (truncation degree 128)

(b) Time Step 25 (truncation degree 32)

(c) Time Step 50 (truncation degree 128)

(d) Time Step 50 (truncation degree 32)

(e) Variance at Time Step 25 (truncation degree 128)

(f) Variance at Time Step 25 (truncation degree 32)

(g) Variance at Time Step 50 (truncation degree 128)

(h) Variance at Time Step 50 (truncation degree 32)

Figure G5: Comparison of model predictions for different models and time steps.

