# OpenReview forum: "SPDEBench: An Extensive Benchmark for Learning Regular and Singular Stochastic PDEs"
_ICLR.cc/2026/Conference — Submitted to ICLR 2026_

### Official Review · Reviewer_R8FC · 2025-10-28

**Soundness:** 3
**Presentation:** 2
**Contribution:** 3
**Rating:** 2
**Confidence:** 4

**Summary:**

The paper introduces SPDEBench, a benchmark suite and dataset collection for machine learning models that approximate stochastic partial differential equations (SPDEs). It includes both nonsingular (e.g., wave, KdV, Navier–Stokes) and singular SPDEs (Φ^4_d, KPZ), with attention to renormalization and noise truncation effects. The benchmark evaluates ML surrogates such as FNO, DeepONet, NSPDE, and DLR-Net, proposing also a modified NSPDE-S model that incorporates renormalization parameters.

**Strengths:**

Novelty: First attempt to provide an extensive, unified dataset for SPDEs (especially singular ones with renormalization).

Technical rigor: Correct implementation of noise processes (cylindrical/Q-Wiener), Wick powers, and renormalization constants — mathematically consistent with Hairer’s regularity structures.

Reproducibility: Datasets, hyperparameters, and code organization (PyTorch + Parquet format) are well described.

Evaluation metrics: Goes beyond RMSE — includes Sobolev norms, correlation, and autocorrelation metrics.

**Weaknesses:**

Real-world significance concern: While SPDEs are mathematically elegant, the chosen benchmark problems (Φ^4_d, KPZ, KdV) are mostly toy or theoretical models rather than real-world SPDEs encountered in climate, finance, or turbulence modeling. The paper claims “physical significance,” but in practice, most examples (especially Φ^4_d models) are of theoretical physics interest, not directly relevant for engineering or applied simulations. No applied domain (e.g., stochastic fluid flow forecasting, atmospheric modeling) is demonstrated, so the “practical impact” remains unclear.

Benchmark representativeness concern: The included SPDEs cover a limited subset. What is missing are hyperbolic SPDEs, reaction-diffusion systems, and stochastic transport equations, which are more common in applied settings. All experiments are on low-dimensional (1D–2D) toy setups with small grid sizes (e.g., 32×32). This limits generalizability. The authors even admit in §5 that “hyperbolic SPDEs are underrepresented.” So, while “extensive” in mathematical diversity, SPDEBench is not truly representative of real physical or industrial systems.

Compared models concern: FNO and DeepONet are standard. NSPDE and DLR-Net are recent but not yet widely used — effectively internal to a small research niche. The others (e.g., NCDE, NRDE, NCDE-FNO) are from rough path theory literature, rarely used for PDEs. Their inclusion feels inconsistent with mainstream PDE ML practice. No comparison to modern operator-learning baselines like UNet-NO, Graph Neural Operators, or Transformers for PDEs (e.g., Galerkin Transformers).

Experimental concern: Noise levels and truncation degrees are somewhat arbitrary; the benchmark lacks connection to any physical noise statistics. Computation is limited to small sample counts (e.g., 1200–10000), not reflective of large-scale scientific computing setups. Scaling analysis (Table 5) focuses only on small network and dataset sizes; conclusions about scaling laws are weak.

Overall: The paper sometimes overstates its importance (“first extensive benchmark,” “physically significant”) without strong evidence of external adoption or real-world scenarios. It would benefit from clearer positioning as a mathematical or synthetic benchmark rather than an applied one.

**Questions:**

See weakness. Please clarify these weak points.

---

> ### Author Response · Authors · 2025-11-22
> **Response**
>
> (1). **Response to "Real-world Significance Concern"**
>
> The primary goal of SPDEBench in its current form is to serve as a controlled testbed for ML-model development, not to directly solve specific applied problems. We’d like to emphasize that the chosen models ($\Phi^4$, KdV, Stochastic Navier-Stokes) have fundamental significance in statistical physics and fluid mechanics, which are mathematically well-understood and encapsulate core challenges like  non-convex potentials ($\Phi^4$), and soliton dynamics with noise (KdV). Success on the benchmarks indicates a model's ability to handle fundamental stochastic and nonlinear phenomena that underpin more complex real-world systems.
>
> **Path to Application**: In the revised manuscript, we will explicitly outline how the challenges identified in our benchmark are directly relevant to applied fields in the discussion paragraph, such that the designed network can be applied to model real-world stochastic system and to study how prediction changes under the spatial-temporal noise.
>
>
>
> (2). **Response to "Benchmark Representativeness Concern"**
>
> Our initial selection focused on establishing a core set of fundamental and represenative SPDEs with significant theoretical and practical relevance, covering parabolic (\Phi^4， KPZ，Navier-Stokes equation), hyperbolic (wave equation), dispersive (KdV equation) and nonlinear SPDEs, where the mathematical theory for simulation is most established. We follow the simulation settings including grid sizes, dimensionality in this research area.  Expanding the benchmark to a broader family of SPDEs will require substantially more computational resources and time effort, and we will emphasise this as a key direction for future community-driven extensions. The current code pipeline is designed to facilitate such expansion and lower the barrier for contributions.
>
> (3). **Response to "Model selection Concern"**
>
> Rough path based models are state-of-the-art models for the SPDE learning in the literature. As their model design explicitly accounts for the structure of stochastic noise and the mild solutions of SPDEs. We don’t include the Graph Neural Network (GNN) based method, as the SPDE data used in our paper is defined on the regular grid, while GNNs primarily handle irregular grid. Following the reviewer’s suggestion, we plan to add a wavelet-based operator as baseline, which allows us to examine whether Fourier bases and wavelet bases for noise representation introduce different inductive biases in neural networks.
>
> (4). **Response to "Experimental Concern"**
>
> **Dataset Size**:  The current sample counts are appropriate for the scope of this benchmark, which is designed for methodological comparison and diagnostic analysis, rather than production-scale training. Within this regime, the relative L^2 errors have shown clear and meaningful differences between models, enabling fair and reproducible comparisons. We will clarify that this is intended for method development, not production-scale training. Moreover, not all ML tasks have access to large-scale datasets; in many SPDE settings, generating high-fidelity samples is computationally expensive, and developing ML models on small- to medium-sized data regimes are also valuable. Our benchmark reflects this realistic constraint.
>
> **Noise Parameters**:  We will provide a theoretical approximation error and visualizations with respect to the noise level in the revised manuscript.
>
> **Scaling Laws**: We will reframe this section to present it as an initial exploratory study that highlights the need for more extensive computational resources to draw definitive scaling laws, and we will emphasize it as an important direction for future community effort.

---

> ### Author Response · Authors · 2025-12-01
> **Additional Experiments to supplement response (3) & (4)**
>
> Dear Reviewer, we have revised the manuscript and added the following experiments to supplement the above response.
>
> - **Model selection concern**
>
> We have added the experiments on wavelet neural operator on the $\Phi^4_1$-H and $\Phi^4_1$-F datasets. We tested the performance for Fourier neural operator (FNO) and wavelet neural operator (WNO) and results are shown in the following table. We will add the training details and the results in the revised paper. We observe that the FNO model significantly and consistently outperforms the WNO model in predicting the dynamic $\Phi_1^4$ equation. For the $\Phi_1^4$-F task: The FNO error (0.134 / 0.148) is approximately 70% lower than the WNO error (0.406 / 0.454).
> For the $\Phi_1^4$-H task: The FNO error (0.141 / 0.157) is approximately 65% lower than the WNO error (0.414 / 0.443). Both models exhibit a slight increase in error when the noise truncation degree increases from J=128 to J=256.
> However, FNO demonstrates a smaller performance degradation, indicating better numerical stability.
>
> **Table 2**: Models' relative $L^2$ error on test set for task $\xi\rightarrow u$ on the dynamic $\Phi_1^4$ with Fourier basis or Haar wavelet basis. Data is generated with $J=128$ or $256$ and $\sigma=1$.
>
> |Model | Dynamic $\Phi_1^4$-F (J=128 / 256)  | Dynamic $\Phi_1^4$-H (J=128 / 256)|
> |------|------|------|
> |FNO|0.134 / 0.148 | 0.141 / 0.157 |
> |WNO| 0.406 / 0.454 | 0.414 / 0.443 |
>
> - **Experimental concern on scaling law**
>
> We have added more baseline models including DeepOnet, FNO, NSPDE, DLR-Net for comparing the scaling law. The results are shown in the following table.
>
> **Table 2**: Relative $L^2$-error for task $\xi\rightarrow u$ on the test set of the dynamic $\Phi_1^4$ model with different training and evaluation sample size and network size. Data is generated with $J=128$ and $\sigma=1$. The sample size refers to the total amount of data utilized for the model development process, including the training and validation sets. $nW$ (or $nD$) denotes $n$ times of the network width (or $n$ times of the network depth). The entry "/" indicates the configuration is inapplicable as the base model depth is $D=1$.
> |Model / Sample Size|1000|2000|3000|5000|10000|
> |----------------------------|------|------|------|------|------|
> |DeepONet|0.840752|0.821926|0.794465|0.738755|0.600467|
> |FNO|0.133919|0.133818|0.132809|0.131829|0.128889|
> |NSPDE|0.018516|0.018482|0.018378|0.018913|0.018185|
> |DLR-Net|0.000795|0.000722|0.000674|0.000579|0.000475|
>
> |Model / Network Size|W*D|(2W)*D|(4W)*D|W*(2D)|W*(D/2)|
> |----------------------------|------|------|------|------|------|
> |DeepONet|0.840752|0.845179|0.842743|0.845750|0.861782|
> |FNO|0.133919|0.066011|0.005871|0.137363|/|
> |NSPDE|0.018516|0.018505|0.018502|0.018674|/|
> |DLR-Net|0.000795|0.000698|0.000475|0.003232|0.000896|
>
> We have two main observations: for all four models, performance improves slightly as the amount of data increases. When the network width expands, FNO demonstrates relatively good scaling capability.
>
> - **Experimental concern on noise parameter**
>
> We have added the theoretical approximation error with respect to the noise truncation degree $J$ in Appendix C.3, which scales as $O(J^{-\alpha+\delta})$. Here, $\alpha\in (0, 2/9)$ denots the index of the Holder-Sobolev space $C^{\alpha}$ to which the solution belongs. $\delta > 0$ can be taken arbitrarily small.
>
> Furthermore, we have plotted the predicted solutions from three models: DLR-Net, NSPDE and FNO on the task of predicting the solution of dynamic $\Phi_1^4$ model with $J=32$ and $J=128$ in Appendix G (Figure G5). We observe that the variance of the ground truth with noise truncation degree $J=128$ is significantly larger  than that with noise truncation degree $J=32$, showing the **significant impact of the noise truncation degree** on the statistics of the solution.

---

### Official Review · Reviewer_vwLn · 2025-10-30

**Soundness:** 2
**Presentation:** 2
**Contribution:** 2
**Rating:** 4
**Confidence:** 2

**Summary:**

This paper presents SPDEBench, a large-scale and unified benchmark for learning stochastic partial differential equations (SPDEs) using machine learning models.
The authors introduce a comprehensive dataset covering both non-singular and singular SPDEs (such as Φ⁴, KPZ, and Navier–Stokes equations), including data generated with renormalization procedures that are essential for handling singular cases. The benchmark provides APIs for dataset generation, multiple evaluation metrics (e.g., RMSE, Sobolev norms, correlation scores), and includes several baseline models such as FNO, DeepONet, NSPDE, and DLR-Net, along with a new variant NSPDE-S that accounts for renormalization constants.
Extensive experiments demonstrate the effectiveness and scalability of the benchmark, showing that DLR-Net and NSPDE achieve state-of-the-art performance across different noise settings. The work emphasizes reproducibility and offers a standardized framework for comparing ML models on SPDE tasks.

**Strengths:**

1. The paper runs extensive experiments and presents results clearly.
2. The proposed benchmark is well designed。

**Weaknesses:**

1. While SPDEBench provides a valuable benchmark for studying stochastic partial differential equations, the work focuses more on system construction and dataset design than on proposing fundamentally new modeling or algorithmic ideas.
2. The paper devotes substantial space to mathematical background and notation, which can obscure the core methodological message and make the main takeaways harder to follow.
3. Although the experiments are extensive and well-organized, they remain largely descriptive. The paper would be strengthened by a clearer analysis of why certain models perform better and what the results imply about the underlying learning dynamics.

**Questions:**

No questions.

---

> ### Author Response · Authors · 2025-11-22
> **Response**
>
> Thank you for your thoughtful and constructive review. We appreciate your recognition of SPDEBench's value as a benchmark and the extensive, well-organized experiments. Your points are well-taken, and we have a plan to revise the manuscript to directly address them, thereby strengthening the paper's impact and clarity.
>
> (1). **Response to Comment on Focus on Benchmarking vs. New Algorithmic Ideas**
>
> We agree with the reviewer that our primary contribution is the benchmark itself, not a single new model. We respectfully argue that in the emerging field of ML for SPDEs, a rigorous, standardized benchmark is a critical and currently missing prerequisite for driving future algorithmic innovation. Our goal is to provide the foundational infrastructure, like what ImageNet did for computer vision or PDEBench for deterministic PDEs. We envision SPDEBench as a community tool that facilitates the development, evaluation, and comparison of ML methods for SPDEs, thereby accelerating scientific progress in SPDE learning.
>
> (2). **Response to Comment on Excessive Mathematical Background**
>
> The mathematical aspects are essential because data construction for SPDEs, especially in the singular case, depends critically on the correctness of the underlying numerical schemes. In particular, our paper shows that noise truncation significantly affects both data generation and comparative model performance—hence the need for careful exposition.
>
> To make the presentation more accessible to the ML community, we provide the minimal preliminaries of SPDEs solution and numerical schemes. The mathematical rigor is important when introducing these key concepts. We have kept the mathematical background for the SPDE brief, e.g., only focusing on the singular case, and leave all the technical details in the appendix. In the revised version, we will further streamline the main text by shortening the description of the renormalization procedure for the $\Phi^4_2$ dataset and moving some technical explanations to the appendix, while keeping enough detail to ensure rigor and reproducibility.
>
> (3). **Response to Comment on Need for Deeper Analysis Beyond Descriptive Results**
>
> Our current results have demonstrated some inductive bias such as: leveraging the regularity feature (e.g., DLR-Net) and (e.g., NSPDE-S) helps to improve the model’s accuracy and robustness. However, the black-box nature of the neural networks makes it hard to explain the underlying learning dynamics in a mathematical rigorous way. We will try our best to find some insights through visualizing the prediction results and added it in the updated version. It will be appreciated if some analyzing tools about the learning dynamics can be shared.

---

> ### Author Response · Authors · 2025-12-01
> **Additional Experiments for Deeper Analysis to supplement Response (3)**
>
> Dear reviewer, we have plotted the predicted solutions from three models: DLR-Net, NSPDE and FNO on the task of predicting the solution of dynamic $\Phi_1^4$ model. The predicted solutions from each model were compared against the ground truth at both an intermediate time step (step 25) and the final time step (step 50). This comparison assessed the discrepancies in their predictions on mean and variance over the input sequences. We will add the curves in the revised paper. From the results, we observe that DLR-Net is the closest to the ground truth in both mean and variance predictions. In mean prediction, FNO and NSPDE show deviations from the ground truth, particularly in regions with larger predicted x-values and x-values near boundaries. Moreover, in variance prediction, the deviations of FNO and NSPDE from the ground truth variance are even more pronounced than the mean predictions.

---

### Official Review · Reviewer_qB9T · 2025-11-01

**Soundness:** 3
**Presentation:** 2
**Contribution:** 2
**Rating:** 4
**Confidence:** 3

**Summary:**

This paper presents SPDEBench, a machine learning benchmark on Stochastic Partial Differential Equations (SPDEs). The authors consider a renormalization procedure when generating these datasets. The paper benchmarks several ML models, including FNO, NSPDE, and DLR-Net, and proposes an upgrade of NSPDE-S.

**Strengths:**

The paper provides one of the first benchmarks that properly handles singular SPDEs with renormalization. The authors demonstrate that their ML surrogates achieve substantial speedups compared to numerical solvers while maintaining reasonable accuracy.

**Weaknesses:**

- The authors said to use metrics beyond RMSE, such as the Sobolev norm, correlation score, and autocorrelation score. However, these metrics are almost entirely absent from the experimental evaluation. The Sobolev norm appears only once in the appendix (Table 21) as a training loss example. The correlation and autocorrelation scores are not reported.
- The selection of baseline models is limited. Many tables in the appendix report results for only FNO and NSPDE. The authors are encouraged to borrow more architectures from literature reviews, for example wavelet-based and transformer-based.
- Overall, the contributions feel limited, as the datasets largely build on prior simulations from works like Neural SPDE and DLR-Net. The new model (NSPDE-S) proposed by the authors is not well-demonstrated and not impressive. Insights provided are also limited.

**Questions:**

Many are mentioned in Weaknesses. Plus, a large portion of their results is in the appendix. The findings from these other important PDE families are not discussed. It is encouraged to discuss findings/takeaways on the equations other than $\Phi$.

---

> ### Author Response · Authors · 2025-11-22
> **Response**
>
> We sincerely thank the reviewer for their thorough review and valuable feedback.
>
> (1) **Response to the Comment on Evaluation Metrics**
>
> Our code repository has included the implementation of these metrics, although we did not report the correlation and auto-correlation scores in some experiments. We agree that in-depth analyses on all these metrics are valuable. Accordingly, we will add Tables in section 4.1 and 4.2 to report the four metrics for comparing the accuracy and scaling law.
>
> (2) **Response to the Comment on Baseline Models**
>
> We selected FNO and NSPDE due to their strong empirical performance on the SPDE datasets. Following the reviewer’s suggestion, we will add more baseline models soon, such as a wavelet-based operator as baseline, which allows us to examine whether different choices of basis for noise representation (e.g., Fourier vs. wavelet) introduce distinct inductive biases in neural networks.
>
> (3) **Response to the Comment on Limited Contributions and Model Demonstration**
>
> We appreciate the opportunity to clarify the scope and contributions of our work. Regarding the dataset and contributions, while SPDEBench builds upon certain existing simulation techniques, its primary contribution lies in providing a systematic benchmarking framework for SPDEs—addressing a critical gap in the community. More specifically, it has the first singular SPDE dataset, which is not available in the literature; the systematic study on the impact of the dataset construction on ML model comparison: e.g., noise truncation levels or the use of renormalization affect the training and test sets, which may lead to the different rankings of ML models.
> Additional evaluation metrics beyond MSE to assess model quality more comprehensively (e.g., Table 21 in Appendix G). A unified open-source framework with thorough evaluation protocols, and multiple noise types to enable reproducible benchmarking. It can be used as the community tool to benchmark the ML models for learning SPDEs. Concerning the NSPDE-S model, we will improve its demonstration soon in the updated version: (a) We will add additional experiments on other test metrics conducted for Points 2 and 3. (b) We will refine the architecture description to explain the computation of the renormalization constant in section I in the Appendix.
>
> We believe these revisions incorporate all your suggestions and substantially improve the clarity and quality of our work. Thank you again for the constructive feedback.

---

> ### Author Response · Authors · 2025-11-30
> **Additional Experimental Results & Revision**
>
> Dear Reviewer, we have added the following experimental results and all these results will be added in the revised paper.
>
> - **Experiments on different evaluation metrics**
>
> **Table 1**: Different test metrics of models on the test set of $\Phi_1^4$ with $J=256$ and $\sigma=0.1$. $L^2$, "Autocor" and "Cor" are abbreviations of the relative $L^2$ error, auto-correlation and correlation. Lower values are better.
>
> |Model | $\xi\rightarrow u$: $L^2$ / $W^{1,2}$-Sobolev / Autocor / Cor  | $(u_0,\xi)\rightarrow u$: $L^2$ / $W^{1,2}$-Sobolev / Autocor / Cor|
> |------|------|------|
> |NCDE|0.099 / 0.966 / 0.796 / 0.090| 0.147 / 0.919 / 1.664 / 0.167|
> |NRDE|0.181 / 0.864 / 1.056 / 0.257| 0.204 / 0.865 / 1.287 / 0.261|
> |NCDE-FNO|0.070 / 0.807 / 0.077 / 0.014| 0.070 / 0.754 / 0.123 / 0.015|
> |DeepOnet|0.174 / 0.857 / 1.162 / 0.229| not applied|
> |FNO|0.027 / 0.608 / 0.082 / 0.009| not applied|
> |NSPDE|0.004 / 0.061 / 0.006 / 0.0006|0.004 / 0.060 / 0.003 / 0.0006|
> |DLR-Net|0.001 / 0.005 / 0.001 / 0.0001| 0.001 / 0.005 / 0.003 / 0.00006|
>
> Among the evaluated baselines, NSPDE and DLR-Net perform comparably, both achieving top-tier results for metrics $L^2$, auto-correlation and correlation. Notably, most models exhibit degraded performance under the $W^{1,2}$-Sobolev norm except DLR-Net. One potential reason is that $W^{1,2}$-Sobolev norm takes the derivatives of the spatio-temporal solution into account, and the calculation of the regularity feature in the DLR-Net regularizes the derivatives while other baselines do not regularize the higher-order derivatives.
> - **Added baseline models for comparing the scaling law**
>
> We have added DeepOnet, FNO as baselines for comparing the scaling law. The results are shown in the following table.
>
> **Table 2**: Relative $L^2$-error for task $\xi\rightarrow u$ on the test set of the dynamic $\Phi_1^4$ model with different training and evaluation sample size and network size. Data is generated with $J=128$ and $\sigma=1$. The sample size refers to the total amount of data utilized for the model development process, including the training and validation sets. $nW$ (or $nD$) denotes $n$ times of the network width (or $n$ times of the network depth). The entry "/" indicates the configuration is inapplicable as the base model depth is $D=1$.
> |Model / Sample Size|1000|2000|3000|5000|10000|
> |----------------------------|------|------|------|------|------|
> |DeepONet|0.840752|0.821926|0.794465|0.738755|0.600467|
> |FNO|0.133919|0.133818|0.132809|0.131829|0.128889|
> |NSPDE|0.018516|0.018482|0.018378|0.018913|0.018185|
> |DLR-Net|0.000795|0.000722|0.000674|0.000579|0.000475|
>
> |Model / Network Size|W*D|(2W)*D|(4W)*D|W*(2D)|W*(D/2)|
> |----------------------------|------|------|------|------|------|
> |DeepONet|0.840752|0.845179|0.842743|0.845750|0.861782|
> |FNO|0.133919|0.066011|0.005871|0.137363|/|
> |NSPDE|0.018516|0.018505|0.018502|0.018674|/|
> |DLR-Net|0.000795|0.000698|0.000475|0.003232|0.000896|
>
> We have two main observations: for all four models, performance improves slightly as the amount of data increases. When the network width expands, FNO demonstrates relatively good scaling capability.
>
> - **Experiment on wavelet neural operator**
>
> We have tested the performance for Fourier neural operator (FNO) and wavelet neural operator (WNO) and results are shown in the following table. We will add the training details and the results in the revised paper. We observe that the FNO model significantly and consistently outperforms the WNO model in predicting the dynamic $\Phi_1^4$ equation. For the $\Phi_1^4$-F task: The FNO error (0.134 / 0.148) is approximately 70% lower than the WNO error (0.406 / 0.454).
> For the $\Phi_1^4$-H task: The FNO error (0.141 / 0.157) is approximately 65% lower than the WNO error (0.414 / 0.443). Both models exhibit a slight increase in error when the noise truncation degree increases from J=128 to J=256.
> However, FNO demonstrates a smaller performance degradation, indicating better numerical stability.
>
> **Table 3**: Models' relative $L^2$ error on test set for task $\xi\rightarrow u$ on the dynamic $\Phi_1^4$ with Fourier basis or Haar wavelet basis. Data is generated with $J=128$ or $256$ and $\sigma=1$.
>
> |Model | Dynamic $\Phi_1^4$-F (J=128 / 256)  | Dynamic $\Phi_1^4$-H (J=128 / 256)|
> |------|------|------|
> |FNO|0.134 / 0.148 | 0.141 / 0.157 |
> |WNO| 0.406 / 0.454 | 0.414 / 0.443 |
>
> - **Architecture description of NSPDE-S**
>
> We incorporate the normalized $a_\epsilon$ as a scaling factor by multiplying it with the latent space variables in the NSPDE model, the code is
> ```javascript
> a_eps = a_eps.unsqueeze(1)
> zs = self.solver(z0, xi, grid)
> gate = torch.sigmoid(torch.abs(a_eps)) * 2
> zs = zs * gate.
> ```
> Here, $a_\epsilon$ is the renormalization constant computed from the input noise term for learning singular SPDE, and the model is named NSPDE-S.

---

### Official Review · Reviewer_Nxs5 · 2025-11-02

**Soundness:** 2
**Presentation:** 3
**Contribution:** 1
**Rating:** 2
**Confidence:** 3

**Summary:**

This paper introduces SPDEBench, which is a benchmark suite and dataset generator specifically designed to unify and advance machine learning research on SPDE.

**Strengths:**

The idea of a unified, extensible benchmark for SPDEs is original in AI for math. Few (if any) other standardized benchmarks exist for this particular domain.

**Weaknesses:**

My major concern is stated in Questions.


Minor comment:
Long-term value of SPDEBench will depend on the extensibility and community adoption. This paper could discuss how to introduce new dataset in the benchmark tool

**Questions:**

My major question to this paper is what the difference with this neurips paper https://papers.neurips.cc/paper_files/paper/2022/file/0a9747136d411fb83f0cf81820d44afb-Paper-Datasets_and_Benchmarks.pdf


While these papers address different types of problems (PDEs and SPDEs), there is a strong analogy from a machine learning benchmark perspective. In this work, the approaches to dataset structuring, benchmarking, and evaluation appear very similar to those in PDEBench. This raises concerns about the novelty and contribution of SPDEBench, as much of the framework seems adapted from previous work.


This is the primary reason I am recommending a reject at this stage. However, if the authors can provide a solid rebuttal that clarify the substantive differences between SPDEBench and PDEBench, and convincingly demonstrates that the contributions of this work are non-trivial and necessary for the SPDE domain, I would be open to reconsidering my rating.

---

> ### Author Response · Authors · 2025-11-22
> **Response**
>
> We thank the reviewer for the constructive feedback and for highlighting the need to clearly articulate the distinction between SPDEBench and PDEBench. We address the concerns below.
>
> (1) **Fundamental Difference between SPDEs and PDEs (Not a Minor Variation)**
>
> - **SPDEs are inherently stochastic**, with solutions that depend on entire noise trajectories, whereas the solution to a PDE is a deterministic function. Intrinsically and distinctively different from deterministic PDE that appear more often in existing ML benchmarks, SPDEs takes into account the inherent randomness of physical processes, and only degenerates to PDEs when the noise coefficient is set to zero. Formally, as described in Section 2.1 and 2.3, an SPDE solution depends not only on time $t$ and spatial location $x$, but also on an entire noise trajectory $\xi$ living in the function space $C([0,T]\times D;\mathbb{R})$. In contrast, PDE solutions are deterministic function mappings. Learning SPDE solutions is therefore substantially more challenging than learning PDE solutions, as the model must handle an additional infinite-dimensional input, i.e., the noise path, which fundamentally increases the complexity of the function mapping associated learning problem.
> - **The importance and significance of SPDEs**: SPDEs form an independent and essential research field with capabilities far beyond PDEs. (1) They explicitly incorporate randomness, enabling the modelling of turbulence (e.g., stochastic NSE), wave propagation in random media (e.g., stochastic wave equation), and quantum field fluctuations (e.g., $\Phi_d^4$). Their solutions are random fields, providing access to rich statistical information (e.g., expectations, covariances, invariant measures) that PDEs cannot offer. (2) Theoretical challenges are also substantially greater: even with smooth coefficients, space–time white noise is a distribution, making SPDE solutions highly irregular. For singular SPDEs, renormalization is required to ensure well-posedness. The fundamental importance of this renormalization theory is underscored by Martin Hairer’s Fields Medal work on the regularity structure.
>
> (2) **Core differences between SPDEBench and PDEBench**
>
> This single structural change, introducing an infinite-dimensional stochastic input alters every component of the SPDE benchmark pipeline:
>
> - **Dataset generation and tasks**: The SPDE dataset is composed of thousands of the solution trajectories corresponding to different noise paths, not a single deterministic solution. The task is noise-to-solution operator learning tasks (please refer to Section 2.3), which is completely different from those in PDEBench.
>
> - **Numerical simulation method**: The simulation of SPDEs requires the representation of stochastic noise and dedicated numerical solvers due to the low regularity of noise trajectory, such as white or colored noise (please refer to Section 2.2 and C.2). For the singular SPDE case, one requires specialized techniques like renormalization to ensure the validate challenges that are absent in the PDEBench.
>
> - **Comprehensive sensitive analysis of numerical schemes**: The datasets in SPDEBench are generated with a meticulous hyperparameter design, encompassing critical aspects such as noise truncation schemes, basis functions for noise representation, noise types, and renormalization. These parameters are paramount for approximation accuracy and provide a rigorous testbed for evaluating the robustness of machine learning surrogate models.
>
> - **Expanded Benchmark Model Suite**: While PDEBench evaluated three core models (U-Net, FNO, and PINN), SPDEBench features an expanded suite of eight distinct machine learning models. This broader selection facilitates a more extensive comparison and a comprehensive ablation study, offering deeper insights into model performance.
>
> - **Evaluation metrics**: we include the test metrics not only MSE, but also compare distributions (e.g., correlation-based distances) and compare the high order derivatives of the solution in Sobolev norm (Please refer to Section G in Appendix).
>
> (3). **Extensibility and Community Adoption**
> We agree that long-term impact depends on extensibility. SPDEBench is designed with a modular interface that allows researchers to easily add new SPDE datasets, models, and evaluation metrics. We provide example demo in Appendix D illustrating data format for existing SPDE types.  Additionally, our code pipeline is flexible to allow users add more datasets,  ML models and evaluation metrics.  Moreover, the entire code base is fully open-source to ensure reproducibility and to encourage broad adoption and community contributions.

---

### Author Response · Authors · 2025-12-02
**Revision**

Dear SAC, AC, and Reviewers，

We thank the SAC, AC, and Reviewers for their careful evaluation of our manuscript, and we deeply appreciate their constructive and insightful feedback. We would like to respectfully emphasise the scientific importance and novelty of SPDEBench. It is among the very first benchmark suites to provide simulated datasets for SPDEs (not just PDEs), notably including the singular SPDEs. It enables systematic, transparent, and reproducible evaluation of ML surrogate models for spatio-temporal systems where SPDEs are standard modelling tools, hence offering a rigorously controlled testbed for benchmarking relevant ML algorithms.

A key contribution of our work is to highlight the critical role of data construction, oftentimes overlooked in ML literature. Indeed, evaluating ML models solely on datasets simulated under a fixed configuration may limit the understanding of their robustness and approximation capabilities. Furthermore, improperly simulated data can undermine model training and evaluation, especially for singular SPDEs. For example, for the singular 1-D KPZ equation, data generated without renormalisation leads to fundamentally incorrect behaviour (see Appendix Figure C2). Model performance is also sensitive to simulation schemes, hyperparameters, and noise representation. SPDEBench directly addresses these issues by providing constructed datasets under various setting of parameter configurations in numerical analyses.

We would also like to clarify that some reviewers’ concerns—particularly regarding “scaling experiments”/ “lack of empirical datasets”—do not, in our opinion fully align with the nature of this domain. We acknowledge the importance of large-scale empirical data in many ML settings, while SPDEBench targets mathematically structured spatio-temporal PDE/SPDE systems, where moderate-sized but high-fidelity simulated data is the norm, and where experiments are often constrained by computational cost due to the underlying physics-based solvers. This is a different but equally important use case, and SPDEBench aims to establish a principled, extendable benchmarking tool upon which future large-scale or empirical efforts can be built.

We hope the AC and SAC will take these domain-specific considerations into account when assessing the significance and long-term impact of our contribution.

**Revision summary**

Following the reviewers’ constructive feedback, we have added substantial new numerical results, which we believe significantly strengthen the manuscript. Below, we summarise the key revisions made, followed by our point-to-point responses to each reviewer. All changes are highlighted in blue in the updated paper.

- We have revised the first and third paragraphs in the introduction and the first sentence in Section 3.1 to further **emphasize the fundamental significance and representativeness of the selected SPDEs**. We also revised Section 5 (discussions and limitations) to outline the **path to application**. (See points (1) and (2) in our response to **Reviewer R8FC**)

- We have revised the **description of NSPDE-S** in Section 3.2 and added Section I in the Appendix to more clearly explain how the renormalization constant is incorporated within the network. (See "Architecture description of NSPDE-S" in our response to **Reviewer qB9T**)

- We have added the visualization on mean and variance of the prediction curves over samples for ML models in appendix G and revised section 4.1 for **deeper analyses of the ML models** (See point (3) in our response to **Reviewer vwLn**) and **non-arbitrary selection on noise level** (See point (4) in our response to **Reviewer R8FC**).

- We have added **evaluation under different metrics** in section 4.2. (See "Experiments on different evaluation metrics" in our response to **Reviewer qB9T**)

-  We added baseline models DeepOnet, FNO in **comparison of scaling law** in Section 4.3. (See "Added baseline models for comparing the scaling law" in our response to **Review qB9T and R8FC**)

- We have added the **theoretical approximation error** with respect to the noise truncation degree $J$ in Appendix C.3. (See point (4) in our response to **Reviewer R8FC**)

- We have added the experiments on **wavelet neural operator** in Appendix F.2 and in the code. (See our response to **Reviewer R8FC and qB9T**)

Sincerely,

Authors

---

### Meta-Review · Area_Chair_c7v3 · 2026-01-07

**Summary:**

I recommend rejection. While reviewers agree that SPDEBench addresses a real gap in benchmarking machine learning methods for SPDEs, the paper does not meet the ICLR bar in terms of contribution and impact. The main concerns are the limited novelty relative to existing benchmarks (especially PDEBench), the unclear practical significance beyond low-dimensional synthetic setups, and the largely descriptive nature of the experimental results. The rebuttal enhances completeness by introducing metrics, baselines, and analyses, but it does not fundamentally improve the contribution.

**Reviewer Concerns:**

Reviewers' concerns about missing evaluation metrics and limited baselines were largely addressed through the addition of correlation/Sobolev metrics, as well as further models (e.g., DeepONet, FNO).
Reviewers' requests for deeper analysis were partially addressed through the addition of visualizations and qualitative mean/variance comparisons. Reviewers' technical concerns regarding noise truncation and scaling were mitigated by incorporating a theoretical discussion and expanded scaling experiments.

Most concerns remain unresolved. Multiple reviewers questioned the novelty of the benchmark relative to prior benchmark-style works, arguing that the overall structure and contribution appear incremental rather than qualitatively new. Concerns about real-world relevance and representativeness remain, in my opinion, as the benchmark focuses on low-dimensional, mathematically canonical SPDEs. More generally, the work does not deliver strong conceptual or methodological insights that would generalize beyond the specific benchmark setup.

**Reviewer Scores:**

The additional experiments and clarifications would likely lead some reviewers with borderline scores to increase their assessment slightly. However, reviewers who initially viewed the contribution as insufficiently novel or impactful would likely remain unconvinced, possibly moving only from a clear rejection to a borderline score. Overall, the distribution of scores would still fall below the acceptance threshold, supporting a rejection decision.

---

### Decision · Program_Chairs · 2026-01-26

Reject